# Aurora A and cortical flows promote polarization and cytokinesis by inducing asymmetric ECT-2 accumulation

Katrina M Longhini, Michael Glotzer*

Department of Molecular Genetics and Cell Biology, University of Chicago, Chicago, United States

**Abstract** In the early *Caenorhabditis elegans* embryo, cell polarization and cytokinesis are interrelated yet distinct processes. Here, we sought to understand a poorly understood aspect of cleavage furrow positioning. Early *C. elegans* embryos deficient in the cytokinetic regulator central-spindlin form furrows, due to an inhibitory activity that depends on aster positioning relative to the polar cortices. Here, we show polar relaxation is associated with depletion of cortical ECT-2, a RhoGEF, specifically at the posterior cortex. Asymmetric ECT-2 accumulation requires intact centrosomes, Aurora A (AIR-1), and myosin-dependent cortical flows. Within a localization competent ECT-2 fragment, we identified three putative phospho-acceptor sites in the PH domain of ECT-2 that render ECT-2 responsive to inhibition by AIR-1. During both polarization and cytokinesis, our results suggest that centrosomal AIR-1 breaks symmetry via ECT-2 phosphorylation; this local inhibition of ECT-2 is amplified by myosin-driven flows that generate regional ECT-2 asymmetry. Together, these mechanisms cooperate to induce polarized assembly of cortical myosin, contributing to both embryo polarization and cytokinesis.

*For correspondence:
mglotzer@uchicago.edu

Competing interest: The authors declare that no competing interests exist.

## Editor's evaluation

Cytokinesis in animals involves a contractile actomyosin ring, which generates the forces needed for cell division. A key factor controlling actomyosin ring function is a protein called ECT2, which is a regulator of the signalling protein RhoA GTPase. How ECT2 gets to the correct cellular location and how it executes its functions remain a mystery, despite extensive work. This work through a set of beautiful and thorough experiments establishes the mechanism of ECT2 intracellular distribution, which involves integration of spatial and biochemical signals all contributing to the fidelity of cell division.

## Introduction

The division of a single animal cell into two daughter cells is facilitated by an actomyosin-based contractile ring that generates the force that constricts the cell membrane and generates the cleavage furrow (*Basant and Glotzer, 2018*). Furrow positioning is dictated by the position of the mitotic spindle (*Rappaport, 1985*). Work in *Caenorhabditis elegans* has shown that distinct parts of the spindle, namely the spindle midzone and the asters, are each sufficient to induce furrowing (*Dechant and Glotzer, 2003*; *Bringmann and Hyman, 2005*). The active form of the small GTPase RhoA (C.e. RHO-1) is necessary for cleavage furrow induction in these and other contexts (*Kishi et al., 1993*; *Jantsch-Plunger et al., 2000*; *Zhou and Zheng, 2013*). An experimentally induced equatorial zone of active RhoA suffices to induce a furrow in cultured human cells (*Wagner and Glotzer, 2016*). Despite

progress in our understanding of cytokinesis, gaps remain in our understanding of the mechanism by which asters spatially regulate actomyosin contractility.

The guanine nucleotide exchange factor (GEF) Ect2 (C.e. ECT-2) is the primary activator of RhoA during cytokinesis (*Tatsumoto et al., 1999*). This Dbl-homology family member is regulated by numerous factors. Prominent among these is the centralspindlin complex which, despite containing a subunit with a RhoGAP domain, plays a significant role in RhoA activation during cytokinesis. Centralspindlin both induces relief of ECT-2 auto-inhibition and promotes Ect2 recruitment to the cortex (*Yüce et al., 2005*; *Zhang and Glotzer, 2015*; *Niiya et al., 2005*; *Zhao and Fang, 2005*; *Chen et al., 2019*; *Su et al., 2011*; *Lee et al., 2018*). Anti-parallel, bundled microtubule plus ends, localized kinases, and a phosphatase conspire to promote centralspindlin-dependent RhoA activation at the cell equator, midway between the two spindle poles (*Wolfe et al., 2009*; *Burkard et al., 2009*; *Yüce et al., 2005*; *Basant et al., 2015*; *Hertz et al., 2016*; *Nishimura and Yonemura, 2006*).

Despite the established role of the spindle midzone in cytokinesis in diverse metazoa, furrow formation can occur in its absence in some cell types. This is well characterized in *C. elegans* embryos where the centralspindlin-dependent and -independent furrows have been separated genetically (*Dechant and Glotzer, 2003*; *Werner et al., 2007*). Analogous phenomena operate in cultured human cells (*Chen et al., 2021*; *Murthy and Wadsworth, 2008*; *Ramkumar et al., 2021*; *Rodrigues et al., 2015*). In early *C. elegans* embryos, centralspindlin functions in parallel with NOP-1 to promote ECT-2-dependent RHO-1 activation (*Morton et al., 2012*; *Tse et al., 2012*; *Fievet et al., 2012*). Though it is non-essential, NOP-1 can promote formation of an incomplete furrow (*Tse et al., 2012*). In certain circumstances, NOP-1 suffices to induce complete furrow ingression (*Canman et al., 2008*; *Loria et al., 2012*; *Zhang and Glotzer, 2015*).

In addition to being genetically separable, the centralspindlin- and NOP-1-dependent pathways can be spatially separated. Whereas the position of the centralspindlin-dependent furrow correlates with the spindle midzone (*Werner et al., 2007*), the position of the NOP-1 dependent furrow anti-correlates with the position of the spindle asters (*Hird and White, 1993*; *Dechant and Glotzer, 2003*; *Werner et al., 2007*; *Tse et al., 2012*). Specifically, spindle asters are associated with inhibition of contractility. This inhibitory activity has generally been ascribed to the astral microtubules and is often referred to as polar relaxation/astral relaxation (*Wolpert, 1960*; *White and Borisy, 1983*; *Dechant and Glotzer, 2003*; *Werner et al., 2007*; *Mangal et al., 2018*; *Chapa-Y-Lazo et al., 2020*). In unperturbed cells, centralspindlin-dependent and -independent furrows align during cytokinesis.

Regional suppression of cortical contractility also occurs during polarization of the *C. elegans* zygote. After fertilization, the sperm-derived centrosome serves as a symmetry-breaking cue that inhibits cortical contractility (*Goldstein and Hird, 1996*; *Cowan and Hyman, 2004*). This reduction in cortical contractility induces cortical flows that advect PAR proteins, such as PAR-3 and PAR-6, to the nascent anterior of the embryo (*Munro et al., 2004*). Centrosomes locally concentrate Aurora A kinase (C.e. AIR-1), which is required for centrosome-directed displacement of the RhoGEF ECT-2 from the posterior cortex during polarization (*Motegi and Sugimoto, 2006*; *Zhao et al., 2019*; *Klinkert et al., 2019*; *Kapoor and Kotak, 2019*). However, the mechanism of action of Aurora A is not known, in particular it is not known whether Aurora A regulates ECT-2 during cytokinesis.

To address these questions we quantitatively examined ECT-2 distribution during polarization and cytokinesis in early *C. elegans* embryos. We show that ECT-2 becomes asymmetrically localized coincident with the early stages of polarity establishment, as it becomes depleted posteriorly and enriched anteriorly. The degree of ECT-2 asymmetry attenuates somewhat as the embryo progresses through mitosis and then increases again during anaphase, under the influence of the asymmetric spindle. As during polarization, ECT-2 asymmetry during anaphase requires intact centrosomes and AIR-1. A C-terminal fragment of ECT-2 largely recapitulates the dynamic localization of ECT-2 during early embryogenesis. A set of putative Aurora A sites have been identified in this fragment that are required for its displacement by centrosomes and Aurora A. While phosphorylation of ECT-2 by Aurora A serves as a symmetry breaking cue, ECT-2 asymmetry also requires myosin-dependent cortical flows. Our results suggest that Aurora A-triggered phosphorylation of ECT-2 initiates cortical flows which redistribute actomyosin and associated factors, including ECT-2, thereby regulating polarization and cytokinesis.

## Results

### Cortex-associated ECT-2 is asymmetric during cytokinesis

To dissect the mechanism by which spindle asters regulate contractility, we first examined the distribution of endogenous ECT-2 during early embryogenesis. Displacement of ECT-2 from the posterior cortex during polarization has been previously described in studies utilizing an ECT-2:GFP transgene. Collectively, these works revealed that centrosomes and/or their associated microtubules induce displacement of ECT-2 during polarization (*Motegi and Sugimoto, 2006*; *Kotak et al., 2016*; *Kapoor and Kotak, 2019*). To further investigate the dynamics of ECT-2 localization, we engineered a mNeonGreen (mNG) tag at the C-terminus of endogenous ECT-2 (*Dickinson et al., 2013*). ECT-2::mNG is well tolerated (>99% embryos hatch).

During both polarization and cytokinesis, we quantified the levels of cortical ECT-2 along the entire perimeter of the embryo (*Figure 1—figure supplement 1*). For each time point, we averaged cortical ECT-2 levels in the anterior and posterior 20% of the embryo, and calculated the ratio of these values as a measure of ECT-2 asymmetry. In the early zygote, the cortical pool of ECT-2 increases and it becomes progressively more asymmetric (*Figure 1B* i–iii). Images acquired during polarity establishment reveal displacement of ECT-2 from a large domain in posterior cortex and a progressive increase in the anterior cortex resulting in highly asymmetric cortical ECT-2 (*Figure 1A*, –480 s [relative to anaphase onset], *Figure 1Bi, ii,vi*). The size of the posterior domain where ECT-2 is displaced, that is, boundary length, represents ~35% of the embryo, measured at the time point of maximal ECT-2 asymmetry. ECT-2 accumulation on the anterior cortex is accompanied by an increase in cortical myosin accumulation (*Figure 1—figure supplement 2*), cortical contractility and pseudocleavage furrow formation, and transient elongation of the embryo (*Figure 1B* ii, v). The pseudocleavage furrow forms at the boundary between the myosin-enriched anterior and the myosin-depleted posterior of the embryo; myosin does not solely accumulate at the furrow.

Following pronuclei meeting and centration, overall cortical ECT-2 levels fall and the embryo partially rounds up (*Figure 1B* iv, v). ECT-2::mNG remains enriched on the anterior cortex during prophase and mitosis, and it remains detectable and slightly increases on the posterior cortex (*Figure 1A*, –240 s, *Figure 1B, i-vi*). Thus, as a consequence of its prior polarization, ECT-2 is asymmetrically localized before the onset of cytokinesis. During anaphase, ECT-2 again becomes preferentially displaced from the posterior ~35% of the embryo, becoming more asymmetric (*Figure 1C*, 60 s, *Figure 1D, i-iii*). As furrow ingression begins, ECT-2 is not significantly enriched on the posterior cortex above its level in the cytoplasm (*Figure 1C* 100 s). A membrane probe, mCherry-PH, which detects PIP2, also exhibits anterior enrichment, though to a lesser degree than ECT-2::mNG and its levels remain stable as anaphase progresses (*Figure 1—figure supplement 3*; *Hirani et al., 2019*). Regional displacement of ECT-2 repeats in subsequent cell cycles; the cortical pool of ECT-2 decreases each cell cycle near spindle poles at anaphase and recovers in the subsequent cell cycles (*Figure 1—figure supplement 4*). Thus, during both polarization and cytokinesis, cortical ECT-2 is highly asymmetric.

### ECT-2 distribution during cytokinesis is regulated by centrosomes

Microtubule-rich asters suppress contractility in the polar regions during cytokinesis (*Werner et al., 2007*). During spindle elongation, a correlation between spindle rocking and ECT-2 displacement is apparent: as the spindle pole more closely approaches one region of the cortex, the ECT-2 on the cortex in that region is preferentially displaced following a short (~10 s) delay (*Figure 2A*).

These results suggest that cortical levels of ECT-2 are modulated by the aster, though they do not distinguish between the role of the centrosome or the astral array of microtubules. To determine whether microtubules are required, embryos were depleted of both alpha- and beta-tubulin and treated with nocodazole to depolymerize microtubules to the greatest extent possible. Despite significant depletion of tubulin and near-complete depolymerization of microtubules (*Figure 2B*, insets), we observed strong displacement of ECT-2 from a broad region of the posterior cortex and the initiation of cortical contractility following mitotic exit (*Figure 2B*). In these embryos, the centrosomes could be identified by the residual signal from mCherry-tubulin, and the posterior displacement was centered around the position of the centrosome. ECT-2 becomes hyper-asymmetric in these embryos as compared to control embryos and the depleted zone comprises a larger region of the embryo (*Figure 2B*), as is also observed in embryos with a diminished, posteriorly positioned spindle (see below).

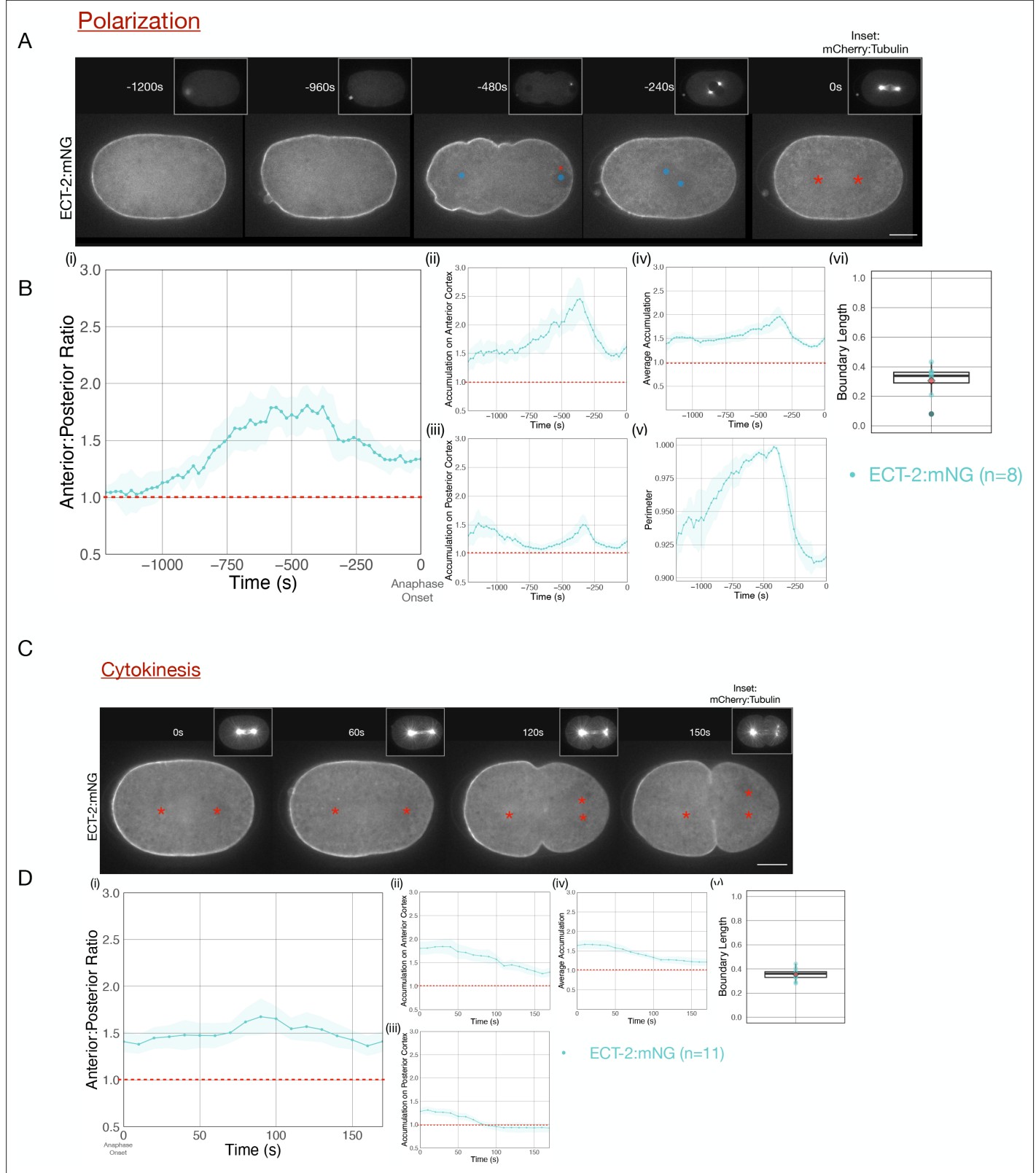

**Figure 1.** Cortical ECT-2 becomes asymmetric during polarization and cytokinesis. (**A**) ECT-2 accumulation is asymmetric during polarization. Selected time points from a representative embryo expressing ECT-2:mNeonGreen (mNG) and mCherry:Tubulin (insets) during polarity establishment. Time zero represents anaphase onset. Blue circles indicate pronuclei and red asterisks label the approximate locations of the centrosomes. All scale bars represent 10 μm. (**B**) Quantification of (**i**) average anterior:posterior ratio of anterior and posterior cortical accumulation, (**ii**) average anterior cortical accumulation,

*Figure 1 continued on next page*

*Figure 1 continued*

(iii) average posterior cortical accumulation, (iv) average cortical accumulation over the entire cortex, and (**v**) normalized perimeter of embryos over time (**s**). n=8; solid line is the average of all embryos measured and the shaded ribbon represents the 95% confidence interval (95 CI). (vi) Box plot of boundary length. Boundary length is the fraction of the embryo cortex with normalized ECT-2 levels below threshold calculated at the time point of the maximum anterior:posterior ratio of ECT-2 for each embryo (see Materials and methods for details). Source data is available (*Figure 1—source data 1*). (**C**) ECT-2 accumulation is asymmetric during cytokinesis. A representative embryo expressing endogenously tagged ECT-2:mNeonGreen (mNG) at the indicated time points during cytokinesis beginning at anaphase onset (0 s). Red asterisks label the approximate locations of the centrosomes. (**D**) Quantification of the (**i**) average anterior:posterior ratio, (ii) average anterior cortical accumulation, (iii) average posterior cortical accumulation, and (iv) average cortical accumulation. n=11; solid line represents the average of all embryos and the shaded ribbon represents the 95% confidence interval (95 CI). (**v**) Box plot of boundary length at the most asymmetric time point of ECT-2 for each embryo. Source data is available (*Figure 1—source data 1*).

The online version of this article includes the following source data and figure supplement(s) for figure 1:

**Source data 1.** This source data table consists of the measurements of ECT-2 accumulation as a function of time.

**Figure supplement 1.** Schematic describing quantitation of ECT-2 cortical accumulation.

**Figure supplement 2.** ECT-2 and cortical myosin accumulate on the anterior cortex during polarization.

**Figure supplement 3.** ECT-2 is more asymmetric than a membrane marker.

**Figure supplement 3—source data 1.** The source data is as described in legend to *Figure 1—source data 1*.

**Figure supplement 4.** ECT-2 accumulates in an asymmetric manner on the cortex in multicellular embryos.

To further examine the role of the centrosome in ECT-2 asymmetry, we depleted the centrosomal scaffold protein SPD-5. Embryos depleted of SPD-5 lack organized arrays of microtubules, though some microtubules assemble in a poorly organized manner (*Hamill et al., 2002*; *Pelletier et al., 2004*). ECT-2 asymmetry is reduced in these embryos both during polarization, as previously reported (*Kapoor and Kotak, 2019*), and during cytokinesis; in particular, significant ECT-2 remains on the posterior cortex (*Figure 2C* iii). Collectively, these studies suggest that ECT-2 asymmetry during anaphase is centrosome-directed.

The evidence presented thus far indicates that centrosomes are the primary source of an activity that induces ECT-2 displacement during anaphase. Although the embryo contains two centrosomes at the time of the first cell division, one in the anterior, the other in the posterior, ECT-2 is preferentially displaced from the posterior cortex. Two, non-mutually exclusive explanations could underlie these differential responses. First, the centrosomes could be functionally distinct as a consequence of the disparate ages and maturity of the embedded centrioles – and assuming that the centrosomes undergo age-dependent stereotyped anterior-posterior positioning in the early embryo (*Yamashita et al., 2007*). Alternatively, the two centrosomes may be functionally equivalent, but, due to the underlying polarity of the embryo, the posterior centrosome is more closely opposed to the posterior cortex. To test the second model, we depleted embryos of PAR-3, to disrupt PAR protein polarity, allowing the centrosomes to be situated equidistant to the cortices on the anterior and posterior regions of the embryo (*Etemad-Moghadam et al., 1995*). Upon anaphase onset, embryos depleted of PAR-3 exhibit displacement of ECT-2 from both the anterior and posterior domains (*Figure 2D*), indicating that both centrosomes are competent to inhibit ECT-2 accumulation. Likewise, ECT-2 asymmetry during cytokinesis is reduced in embryos depleted of PAR-2 (*Figure 2—figure supplement 1*). To modulate spindle positioning without disrupting polarity, we depleted the Gα proteins that function redundantly to promote anaphase spindle elongation (*Gotta and Ahringer, 2001*). In Gα-depleted embryos, ECT-2 accumulation is asymmetric during M-phase, but upon mitotic exit, ECT-2 dissociates slowly from the anterior cortex, as in the wild-type, yet its dissociation from the posterior cortex is significantly attenuated (*Figure 2—figure supplement 2*), due to the reduced spindle elongation toward the posterior cortex. As a consequence, rather than ECT-2 accumulation becoming more asymmetric as anaphase progresses as in the wild-type, it becomes nearly symmetric. Collectively, these data indicate that, during cytokinesis, displacement of ECT-2 from the cortex is highly sensitive to centrosome-cortex separation.

## Cortical flows contribute to asymmetric ECT-2 during polarization and cytokinesis

During polarization, the sperm-derived centrosomes that trigger asymmetric accumulation of ECT-2 are small, while ECT-2 is broadly displaced from the posterior region of the cortex, comprising ~35%

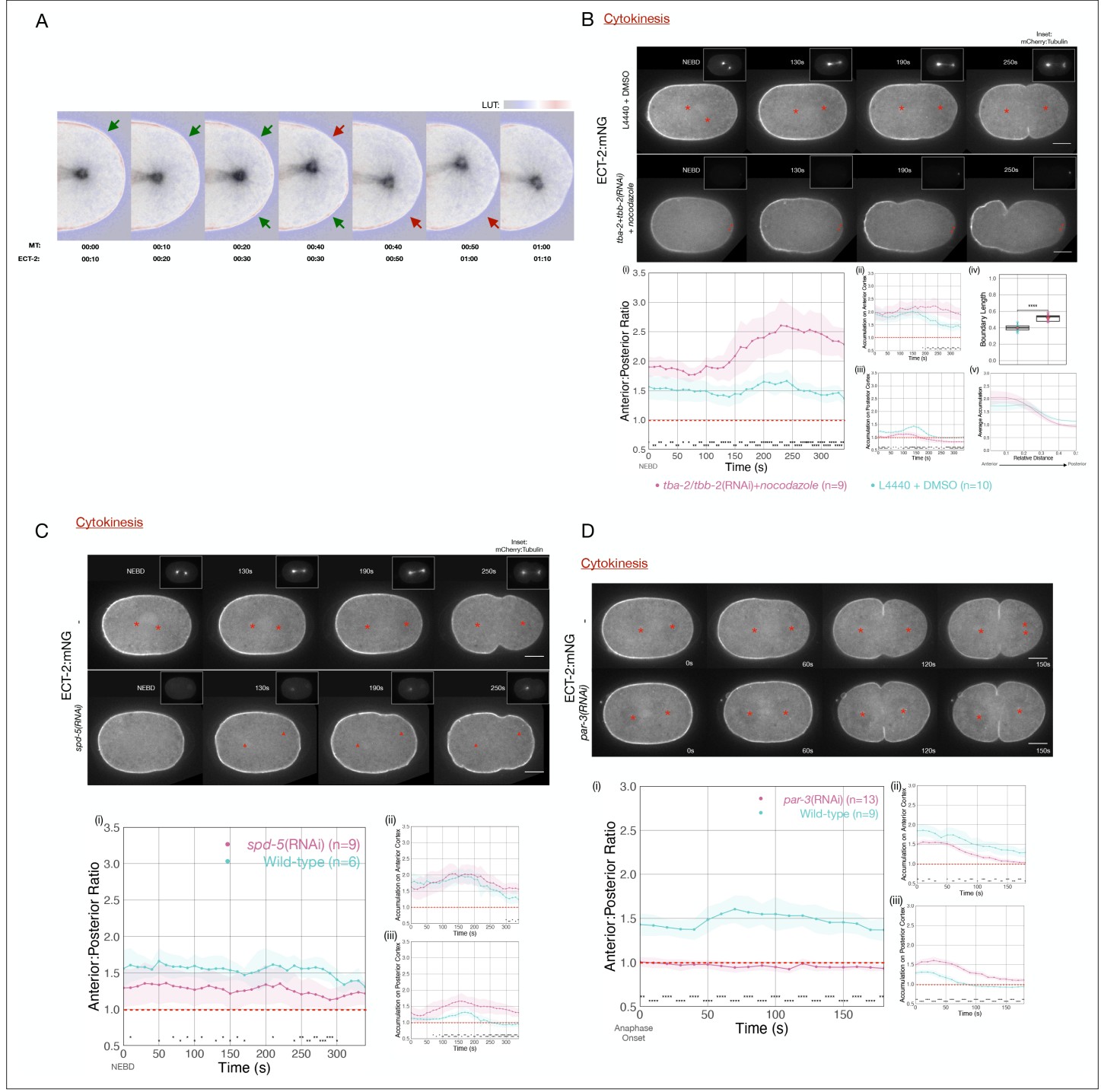

**Figure 2.** ECT-2 asymmetry is responsive to the position of the spindle. (**A**) A close-up view of the localization of endogenously tagged ECT-2 and tubulin during first cleavage. Images from a time-lapse series show the distribution of ECT-2:mNG and mCherry:Tubulin. Composite images are shown: ECT-2 shown in using the displayed lookup table (LUT); the tubulin signal is displayed with inverted grayscale. In the overlays, the tubulin image is advanced by 10 s relative to ECT-2. Arrows highlight two sites where local accumulation of ECT-2 declines ~10 s after the spindle approaches, as indicated by the color change in the arrows from green to red. (**B**) ECT-2:mNG localization in embryos under control [L4440 + DMSO] or experimental [*tba-2+tbb-2*(RNAi) + 50 µg/mL nocodazole] conditions. The images are stills from a time lapse showing ECT-2:mNG localization while the insets are mCherry:Tubulin images acquired concurrently. Asterisks label the approximate locations of the asters. (i–iii) The graphs measure the anterior:posterior accumulation ratio, anterior accumulation, and posterior accumulation of ECT-2 relative to the cytoplasm over time starting at NEBD with 95% confidence interval (CI) labeled as shaded ribbons. (iv) The box plot graphs posterior boundary length of ECT-2 at the maximum polarized time point while (**v**) the line graph plots average accumulation at the same time point as a function of cortical position relative to the anterior of the embryo. For

*Figure 2 continued on next page*

*Figure 2 continued*

this and all subsequent figures, asterisks on graphs indicate statistically significant differences between wild-type and experimental conditions at each indicated time point (*p<0.05, **p<0.01,***p<0.001, ****p<0.0001). Where multiple comparisons are present, the asterisk color reflects the experimental condition compared to wild-type. All scale bars represent 10 µm. Source data is available (*Figure 2—source data 1*). (**C**) ECT-2 localization in an *spd-5*(RNAi) and wild-type embryos beginning at NEBD. The insets show images of mCherry:Tubulin acquired at the same time points. The red asterisks label the approximate locations of the asters while the red triangles label the approximate locations of the compromised asters. The graphs measure the anterior:posterior accumulation ratio, anterior accumulation, and posterior accumulation of ECT-2 relative to the cytoplasm over time. Source data is available (*Figure 2—source data 1*). (**D**) ECT-2 localization in a *par-3*(RNAi) and wild-type embryos with images starting at anaphase onset. Red asterisks label the approximate locations of the asters. The graphs measure the anterior:posterior accumulation ratio, anterior accumulation, and posterior accumulation of ECT-2 relative to the cytoplasm over time. Source data is available (*Figure 2—source data 1*).

The online version of this article includes the following source data and figure supplement(s) for figure 2:

**Source data 1.** The source data is as described in legend to *Figure 1—source data 1*.

**Figure supplement 1.** ECT-2 localization in *par-2*(RNAi) and L4440-treated embryos with images starting at anaphase onset.

**Figure supplement 1—source data 1.** The source data is as described in legend to *Figure 1—source data 1*.

**Figure supplement 2.** ECT-2 localization in *gα*-depleted and wild-type embryos with the representative stills starting at anaphase onset.

**Figure supplement 2—source data 1.** The source data is as described in legend to *Figure 1—source data 1*.

of the embryo cortex (*Figure 1*, –480 s). Likewise, during cytokinesis, the zone of ECT-2 depletion is nearly as broad, and further increased when microtubules are depolymerized (*Figure 2B*). These observations suggest that ECT-2 asymmetry may involve processes that amplify the local centrosomal cue.

Anterior-directed, actomyosin-dependent, cortical flows of yolk granules and anterior PAR proteins are associated with both polarization and cytokinesis (*Hird and White, 1993*; *Munro et al., 2004*). To test whether cortical flows contribute to the displacement of ECT-2 from the posterior cortex during polarization, we examined ECT-2:mNG distribution in *nop-1* mutant embryos which have attenuated cortical flows during polarization (*Rose et al., 1995*). In such embryos, ECT-2:mNG accumulates asymmetrically, suggesting basal flow rates suffice (*Figure 3A*).

To more strongly disrupt cortical flows during polarization, we depleted non-muscle myosin, NMY-2, in embryos expressing ECT-2:mNG (*Munro et al., 2004*). In embryos with reduced myosin-dependent flows, ECT-2:mNG asymmetry is highly attenuated, though a local inhibitory effect on ECT-2:mNG accumulation in the immediate vicinity of the centrosome is detectable; these results are consistent with an earlier report (*Motegi and Sugimoto, 2006*; *Figure 3A* and *Figure 3—figure supplement 1*). These results suggest that cortical flows amplify the local, centrosome-induced reduction in cortical ECT-2 accumulation.

During cytokinesis, ECT-2 activation requires two factors that function in parallel to promote RHO-1 activation, the centralspindlin subunit CYK-4 and NOP-1 (*Tse et al., 2012*). Redistribution of ECT-2 from the posterior cortex could reflect the displacement of ECT-2 in complex with either one or both of these activators. To test each of these possibilities, we examined the distribution of ECT-2 during cytokinesis in embryos in which the functions of NOP-1 and CYK-4 are reduced, either alone or in combination. ECT-2 displacement and asymmetric accumulation was not abolished by these perturbations (*Figure 3B*). Thus, basal levels of RHO-1 activation suffice to promote displacement of ECT-2 from the posterior cortex during anaphase.

Enhanced cortical flows have been reported during cytokinesis in embryos with attenuated astral microtubules, such as nocodazole-treated embryos (*Figure 2B*; *Hird and White, 1993*), embryos with posteriorly positioning spindles due to persistent katanin activity (*Werner et al., 2007*) or depletion of ZYG-9. These perturbations result in suppression of cortical contractility in the posterior cortex and enhancement of contractility in the anterior cortex. These changes induce enhanced cortical flows of myosin toward the anterior that drive formation of an ectopic furrow in the anterior at the boundary between regions of high and low myosin accumulation (*Figure 3C*). Posteriorly positioned spindles also induce a centralspindlin-dependent furrow coincident with the spindle midzone (*Werner et al., 2007*). To quantitatively characterize these flows, we assayed the movement of myosin (NMY-2) punctate in wild-type and ZYG-9-depleted embryos. As compared to control embryos, ZYG-9-depleted embryos contain more cortical myosin foci which travel more than 1 µm, and these foci also move faster and travel further (*Figure 3—figure supplement 3*). To test whether enhanced cortical flows

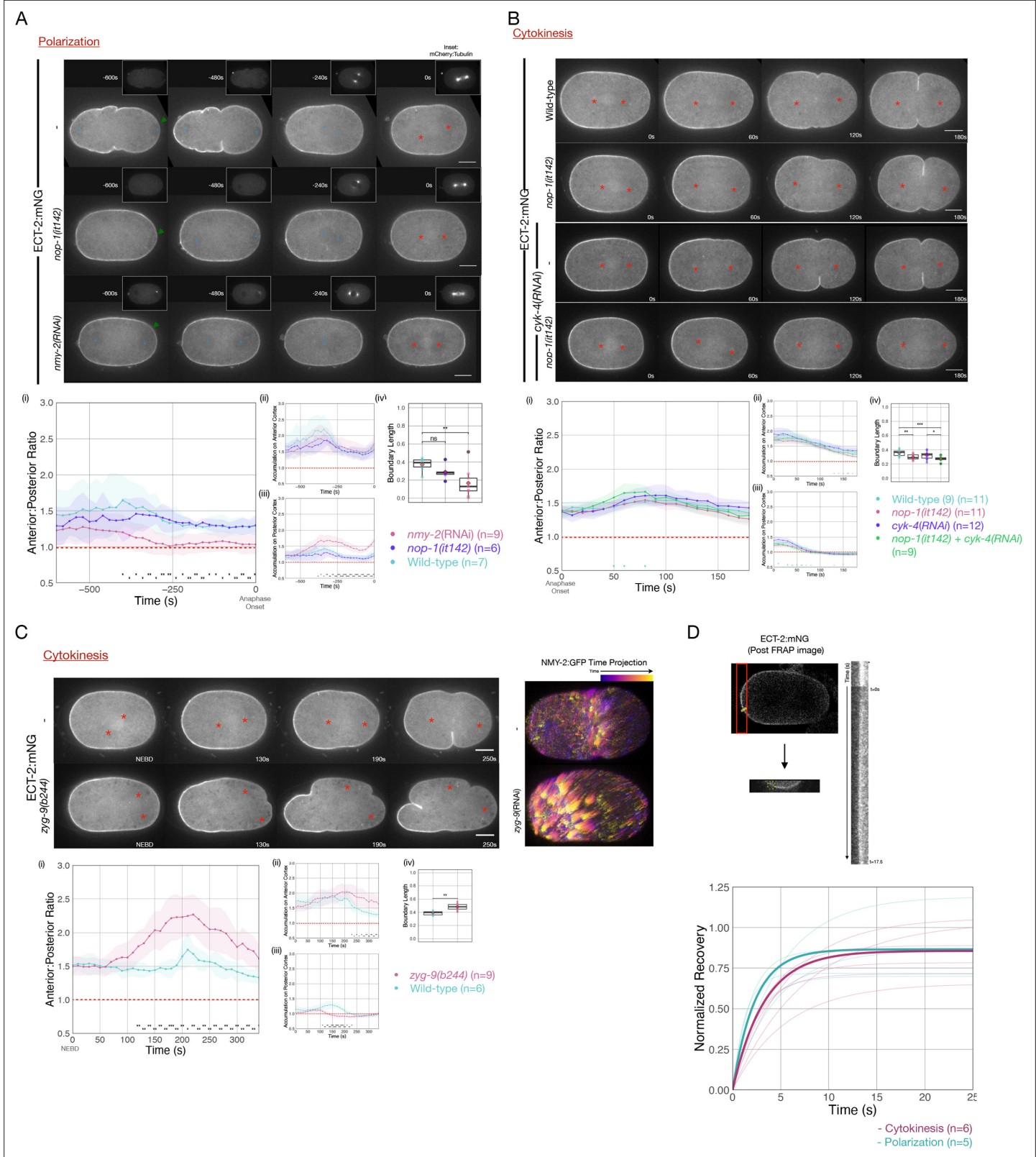

**Figure 3.** Cortical flows contribute to asymmetric cortical accumulation of ECT-2. (**A**) Asymmetric accumulation of ECT-2 during polarization is reduced when cortical flows are reduced. Images from a time-lapse series show ECT-2 localization during polarization in a wild-type, *nmy-2*(RNAi), or *nop-1(it142)* embryo. The pronuclei are labeled with a blue circle and the asterisks label the approximate locations of the asters. Green arrowheads depict local reduction in ECT-2 as the centrosome is opposed to the posterior cortex. Graphs show the anterior:posterior accumulation ratio, anterior accumulation,

*Figure 3 continued on next page*

*Figure 3 continued*

and posterior accumulation of ECT-2 over time. The box plot depicts the posterior ECT-2 boundary at the time point of maximal asymmetry. Source data is available (*Figure 3—source data 1*). (**B**) Asymmetric ECT-2 accumulation is independent of its activators during cytokinesis. Images are stills from time-lapse movies depicting endogenous ECT-2 localization in wild-type, *nop-1(it142)*, *cyk-4*(RNAi), and *nop-1(it142); cyk-4*(RNAi) embryos with time 0 as anaphase onset. Asterisks label the approximate locations of the asters. Graphs measure ECT-2 accumulation relative to average cytoplasm intensity. The boundary length of ECT-2 accumulation was measured at the most asymmetrical time point. Source data is available (*Figure 3—source data 1*). (**C**) The localization of endogenously tagged ECT-2 during cytokinesis is hyper-asymmetric in *zyg-9(b244)* embryos. Images from a time-lapse series show the distribution of ECT-2:mNG in either wild-type or *zyg-9(b244)*. *zyg-9* loss of function results in a posteriorly localized small transverse spindle. Times shown are relative to NEBD which is approximately ~125 s prior to anaphase onset. Red asterisks label the approximate locations of the asters. The NMY-2:GFP images depict the time-dependent changes in cortical myosin in wild-type and *zyg-9*(RNAi) embryos. The images shown are maximum intensity projections of ~25 time points acquired over ~50 s during early anaphase. Each time point is itself a maximum intensity projection of four images separated by 1 μm in Z. Graphs measure ECT-2 accumulation relative to average cytoplasm intensity. Source data is available (*Figure 3—source data 1*). (**D**) Cortical ECT-2 exchanges rapidly. Fluorescence recovery after photobleaching (FRAP) recovery of ECT-2 at the anterior cortex during cytokinesis and on the cortex during polarization. A region of the cortex was imaged and a small ROI was bleached. Twenty-five images were taken at 100 ms intervals before the region was bleached followed by 175 images at 100 ms intervals to document the recovery. The kymograph qualitatively shows the FRAP recovery and the graph shows fitted curves of the recovery of each embryo measured. Recovery was fitted to $y=a*(1-\exp(-b*x))$. The bold lines are the averages of the embryos at either polarization or cytokinesis as indicated ($T_{1/2}$ ~ 3 s). Source data is available (*Figure 3—source data 1*).

The online version of this article includes the following source data and figure supplement(s) for figure 3:

**Source data 1.** This file contains the source data for *Figure 3A–C* is as described in legend to *Figure 1—source data 1*.

**Figure supplement 1.** Asymmetric ECT-2 accumulation is reduced in embryos partially depleted of NMY-2.

**Figure supplement 1—source data 1.** The source data is as described in legend to *Figure 1—source data 1*.

**Figure supplement 2.** ECT-2 asymmetry does not require dynein heavy chain function.

**Figure supplement 2—source data 1.** The source data is as described in legend to *Figure 1—source data 1*.

**Figure supplement 3.** NMY-2:GFP particles were tracked post anaphase onset in wild-type and ZYG-9-depleted embryos using TrackMate in Fiji.

**Figure supplement 3—source data 1.** This source data table contains the velocities and distance traveled of each myosin particle tracked.

**Figure supplement 4.** Fluorescence recovery after photobleaching (FRAP) recovery of the anterior and posterior furrows in an ECT-2:mNG *zyg-9*(RNAi) embryo.

**Figure supplement 4—source data 1.** This source data table includes the individual and average a and b coefficients used to plot the zyg-9(RNAi) fluorescence recovery after photobleaching (FRAP) data.

influence ECT-2 localization, we examined ECT-2::mNG in *zyg-9(b244ts)* embryos. Under these conditions, during anaphase, ECT-2 becomes hyper-asymmetric with a stronger reduction in the posterior cortical pool of ECT-2, an increase in the anterior cortical pool, and an expansion of the posterior domain depleted of ECT-2 (*Figure 3C*). As expected, the two poles of the small spindle appear to suppress ECT-2 accumulation equally. We conclude that cortical flows contribute to ECT-2 asymmetry.

Polar relaxation was proposed to result from dynein-mediated stripping of cortical myosin that generates defects in the cortical network, resulting in anterior-directed flows, contributing to polar relaxation (*Chapa-Y-Lazo et al., 2020*). However, asymmetric ECT-2 accumulation is not abrogated by depletion the heavy chain of cytoplasmic dynein, DHC-1. To the contrary, DHC-1 depletion, which would be predicted to compromise dynein-mediated stripping of cortical myosin, causes enhanced ECT-2 asymmetry, akin to the situation in embryos in which microtubule assembly is strongly suppressed, presumably due to the close association of the centrosomes with the posterior cortex (*Figure 3— figure supplement 2*). Thus, not only are cortical flows required for asymmetric ECT-2, increases or decreases in cortical flows enhance or attenuate the extent of ECT-2 asymmetry, respectively.

To understand how cortical flows promote the asymmetric accumulation of ECT-2, we sought to determine whether ECT-2 itself undergoes long range flows or if ECT-2 dynamically interacts with components that undergo such flows. To that end, we performed fluorescence recovery after photobleaching (FRAP) assays on ECT-2:mNG to measure the dynamics of its cortical association. Following bleaching, ECT-2:mNG rapidly recovers with a $t_{1/2}$ of ~3 s; with no obvious difference detected between polarization and cytokinesis. We conclude that ECT-2 asymmetry is likely to reflect the dynamic association of ECT-2 with more stably associated cortical component(s) that undergo cortical flow (*Figure 3D*).

Although much of ECT-2 is highly dynamic, we found one context in which ECT-2 associates more stably with the cortex. In ZYG-9-deficient embryos, despite the close proximity of centrosomes,

ECT-2:mNG accumulates on the furrow that forms at the posterior pole, in a centralspindlin-dependent manner (*Werner et al., 2007*). FRAP analysis of the posterior and anterior furrows indicates that in both cases the mobile fraction of ECT-2 recovers quickly, but only ~50% of ECT-2 on the posterior furrow recovers, whereas the anterior furrow more fully recovers (*Figure 3—figure supplement 4*).

## AIR-1, an Aurora A kinase, regulates ECT-2 asymmetry during cytokinesis

To further examine the origin of ECT-2 asymmetry, we examined the impact of Aurora A kinase (AIR-1) on ECT-2 distribution during cytokinesis. AIR-1 has been shown to play a central role in embryo polarization (*Kapoor and Kotak, 2019*; *Zhao et al., 2019*; *Klinkert et al., 2019*). Although AIR-1 functions in polarity establishment, AIR-1-depleted embryos are not unpolarized. Rather, in most such embryos, due the existence of a parallel, pathway (*Motegi et al., 2011*), posterior PAR proteins become enriched at both the anterior and the posterior cortex, and cortical flows emanate from both poles toward the equator (*Kapoor and Kotak, 2019*; *Zhao et al., 2019*; *Klinkert et al., 2019*; *Reich et al., 2019*). As a consequence, depletion of AIR-1 by RNAi reduces, but does not abolish, ECT-2 asymmetry during cytokinesis (*Figure 4A*). The residual asymmetry in ECT-2 accumulation in AIR-1-depleted embryos is further reduced by co-depletion of PAR-2 (*Figure 4—figure supplement 2*).

The reduction in cytokinetic ECT-2 asymmetry upon depletion of AIR-1 may reflect a direct role of AIR-1 in ECT-2 localization or it may result from the small spindles that assemble in such embryos (*Hannak et al., 2001*), which, in turn, cause the centrosomes to lie more distal from the cortex. To distinguish between these possibilities, we depleted AIR-1 in embryos defective for ZYG-9 function. Although the spindle assembles close to the posterior pole, it does not induce the pronounced ECT-2 asymmetry observed in the presence of AIR-1 (*Figure 4C*). We conclude that AIR-1 promotes displacement of ECT-2 from the cortex during anaphase.

Given that AIR-1 has multiple functions in centrosome maturation, microtubule stability, and embryo polarization, the effect of AIR-1 depletion on ECT-2 accumulation might result from indirect effects at an earlier stage of the cell cycle (*Hannak et al., 2001*; *Zhao et al., 2019*). To rule out these possibilities, we used a chemical inhibitor of AIR-1, MLN8237 (*Sumiyoshi et al., 2015*). Treatment of embryos with MLN8237 at metaphase resulted in a rapid enhancement in cortical ECT-2 accumulation during cytokinesis, indicating that AIR-1 functions during anaphase to promote ECT-2 displacement from the cortex (*Figure 4—figure supplement 1*).

Some, but not all, of AIR-1 functions involve its binding partner, TPXL-1, an ortholog of Tpx2 (*Ozlü et al., 2005*). Tpx2 is a direct activator of Aurora A kinase activity (*Zorba et al., 2014*). Furthermore, TPXL-1 prominently decorates astral microtubules (*Ozlü et al., 2005*), which are positioned where they could conceivably modulate cortical ECT-2 accumulation. Indeed, TPXL-1 has been proposed to promote the displacement of contractile proteins from the anterior cortex of the early *C. elegans* embryo (*Mangal et al., 2018*). To investigate whether an AIR-1/TPXL-1 complex plays a role in ECT-2 displacement during cytokinesis, we depleted TPXL-1 in embryos expressing ECT-2::mNG. Overall cortical ECT-2 accumulation is enhanced by TPXL-1 depletion, though the degree of ECT-2 asymmetry is unaffected (*Figure 4B*). TPXL-1 depletion results in a number of phenotypes, including reduced spindle length during metaphase (*Lewellyn et al., 2010*), which, like AIR-1 depletion, might indirectly impact ECT-2 accumulation by increasing the distance between spindle pole and the cortex. To distinguish between these possibilities, we examined ECT-2 localization in embryos deficient in both ZYG-9 and TPXL-1 so that the attenuated spindle assembles close to the embryo posterior. Under these conditions, we observed robust depletion of ECT-2 at the posterior pole in *zyg-9(b244)* embryos depleted of TPXL-1, but not AIR-1 (*Figure 4C*). We conclude that while AIR-1 is a major regulator of the asymmetric accumulation of ECT-2, the TPXL-1/AIR-1 complex does not play a central role in this process.

To further explore the function of AIR-1 in ECT-2 regulation, we sought to enhance AIR-1 activity. To that end, we depleted PPH-6 or SAPS-1, the two major subunits of the protein phosphatase 6 (*Afshar et al., 2010*). PPH-6 interacts with AIR-1 and is implicated in dephosphorylation of the activation loop of AIR-1 (*Kotak et al., 2016*). Prior work revealed that depletion of either PPH-6 or SAPS-1 blocks pseudocleavage, reduces the accumulation of cortical myosin, and attenuates the forces that drive spindle elongation (*Afshar et al., 2010*) and more recent work demonstrates that AIR-1 is epistatic to PPH-6 (*Kotak et al., 2016*). As expected for a negative regulator of AIR-1, SAPS-1 depletion results

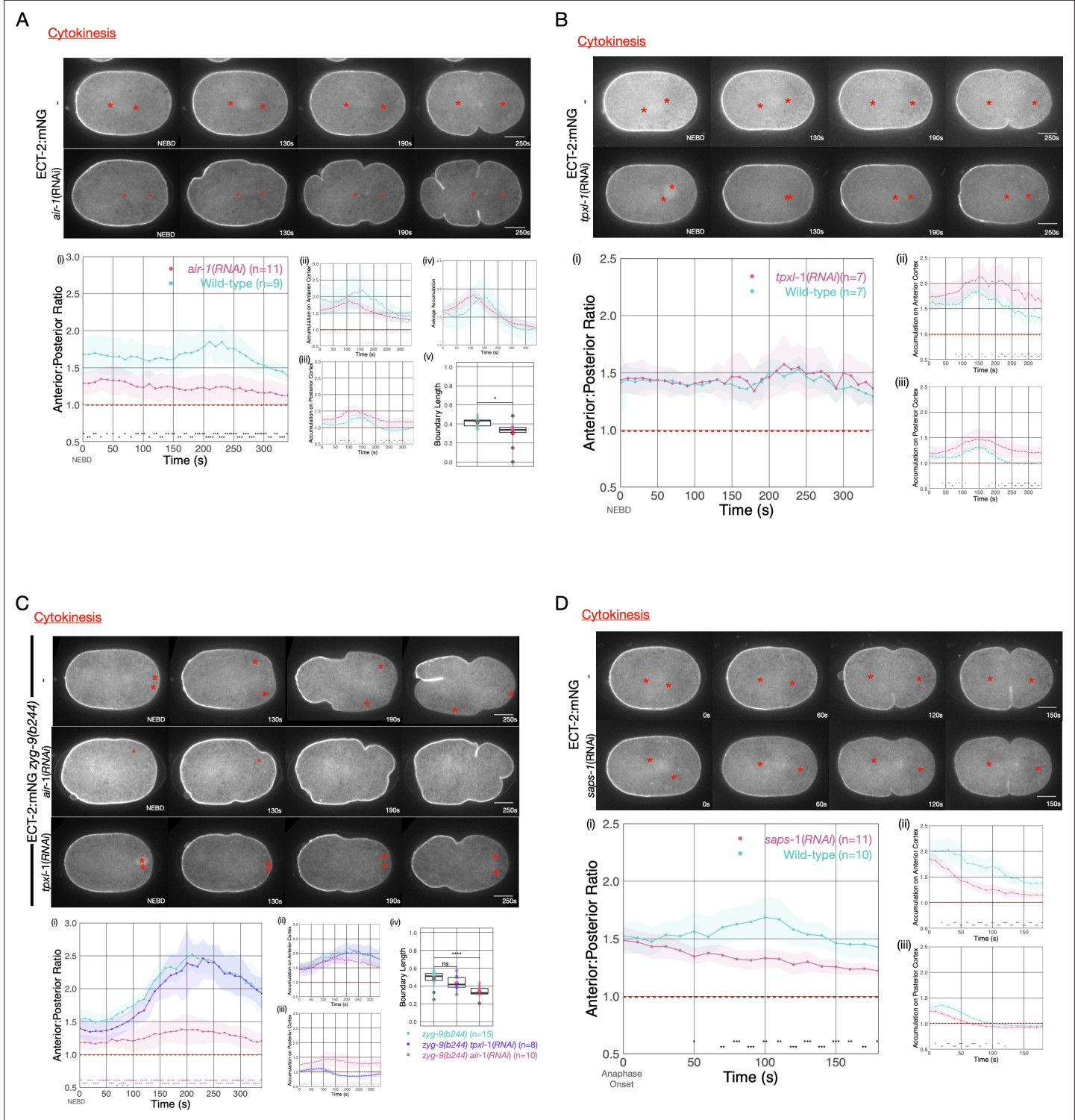

**Figure 4.** Asymmetric ECT-2 accumulation requires AIR-1. (**A**) AIR-1 promotes asymmetric cortical accumulation of ECT-2. Images from a time-lapse series show the localization of endogenously tagged ECT-2 during cytokinesis in either wild-type or embryos depleted of Aurora A kinase, AIR-1, by RNAi. The red asterisks mark the approximate location of the asters while the red triangles mark the center of the unorganized asters. Quantitation of the accumulation of cortical ECT-2:mNG. Note that NEBD is delayed in AIR-1-depleted embryos resulting in contractility at an earlier time point (*Hachet et al., 2007*; *Portier et al., 2007*). Source data is available (*Figure 4—source data 1*). (**B**) TPXL-1 contributes to inhibition of ECT-2 accumulation. Images from a time-lapse series show the accumulation of ECT-2:mNG in either a wild-type or *tpxl-1*(RNAi) embryo. Graphs quantify the cortical accumulation of ECT-2 relative to cytoplasm. Source data is available (*Figure 4—source data 1*). (**C**) AIR-1, but not TPXL-1, is required for inhibition of

*Figure 4 continued on next page*

*Figure 4 continued*

ECT-2 accumulation. ECT-2 was imaged in a *zyg-9(b244)* background in control embryos or embryos depleted of AIR-1 or TPXL-1. Images are shown from a time-lapse acquisition of a representative embryo with NEBD set as time 0. The graphs depict ECT-2 accumulation relative to the cytoplasm and the box plot quantifies the boundary length of ECT-2 inhibition. Note that NEBD is delayed in AIR-1-depleted embryos resulting in contractility at an earlier time point. Source data is available (*Figure 4—source data 1*). (**D**) ECT-2 cortical accumulation is reduced when AIR-1 activity is increased by *saps-1*(RNAi). Images from a time-lapse series show the accumulation of ECT-2:mNG in either a wild-type or *saps-1*(RNAi) embryo with anaphase onset as time 0. Quantitation shows ECT-2 accumulation over time. Source data is available (*Figure 4—source data 1*).

The online version of this article includes the following source data and figure supplement(s) for figure 4:

**Source data 1.** The source data is as described in legend to *Figure 1—source data 1*.

**Figure supplement 1.** Addition of the AIR-1 inhibitor, MLN8237, causes an acute increase in cortical levels of ECT-2:mNG.

**Figure supplement 1—source data 1.** The source data is as described in legend to *Figure 1—source data 1*.

**Figure supplement 2.** Cortical asymmetry of ECT-2 is lost in *air-1 + par-2*(RNAi) embryos.

**Figure supplement 2—source data 1.** The source data is as described in legend to *Figure 1—source data 1*.

in an overall reduction in cortical ECT-2 (*Figure 4D*). Nevertheless, the cortical pool of ECT-2 remains somewhat asymmetric in SAPS-1-depleted embryos. These results indicate that AIR-1 activity inhibits cortical association of ECT-2.

## The C-terminus of ECT-2 recapitulates the asymmetric distribution of ECT-2

To gain insight into the mechanism by which AIR-1 regulates ECT-2 accumulation, we sought to identify a region of ECT-2 that is sufficient to recapitulate the asymmetric accumulation of ECT-2. As previously shown, a C-terminal fragment of ECT-2 that extends from the N-terminus of the DH domain to the C-terminus exhibits cortical accumulation (*Chan and Nance, 2013*). Further deletion analysis indicates that cortical accumulation is independent of the majority of the DH domain, but required both the PH domain and a portion of the region C-terminal to the PH domain, but not its entirety (ECT-2$^C$) (*Figure 5A*). This fragment can localize to the cortex of embryos depleted of endogenous ECT-2 (*Figure 5—figure supplement 1*); although it does not accumulate asymmetrically under these conditions. The accumulation pattern of ECT-2$^C$ is similar to that of endogenous ECT-2 during both polarization and cytokinesis (*Figure 5D and E*). This ECT-2 fragment is also recruited to the posterior, centralspindlin-dependent furrow that forms in ZYG-9-depleted embryos (not shown). AIR-1 depletion resulted in a reduction in the degree of its asymmetric accumulation (*Figure 5—figure supplement 2*). These data indicate that this C-terminal fragment of ECT-2 is sufficient to respond to the inhibitory signal from the centrosomes, and that the N-terminal regulatory domain and the DH domain of ECT-2 are dispensable for asymmetric ECT-2, and its recruitment by centralspindlin.

## Putative phosphorylation sites in the PH domain impact cortical accumulation of ECT-2

As AIR-1 is required for displacement of ECT-2 from the posterior cortex, we examined whether ECT-2 contains putative AIR-1 phosphorylation sites that could regulate its association with the cortex. Using a minimal Aurora consensus site [(K/R)|(K/R)(S/T)] (i.e. at least one basic residue, one to two residues N-terminal to an S/T residue) (*Meraldi et al., 2004*), we identified five putative sites in ECT-2$^C$. Two putative sites were located in an Alpha Fold predicted loop in the PH domain (*Jumper et al., 2021*), as was an additional serine residue with a near fit to the consensus sequence (RHAS$^{643}$) (*Figure 5B and C*). We generated GFP-tagged transgenes in which either all six sites or the three sites in the PH domain loop were mutated to alanine, a non-phosphorylatable residue. As compared to ECT-2$^C$, both variants with alanine substitutions exhibited increased cortical accumulation (*Figure 5D and E*). Nevertheless, these variants accumulated in an asymmetric manner. ECT-2$^C$ asymmetry temporally correlated with anteriorly directed cortical flows (*Figure 5D and E*), raising the possibility that asymmetric accumulation of endogenous ECT-2 drives flows that cause asymmetry of the transgene, irrespective of its phosphorylation status.

To test whether phosphorylation regulates the displacement of ECT-2 derivatives during polarization, we depleted non-muscle myosin, NMY-2, in embryos expressing GFP:ECT-2$^C$ or GFP:ECT-2$^{C\_3A}$. In the absence of myosin-dependent flows, the degree of GFP:ECT-2$^C$ asymmetry is dramatically

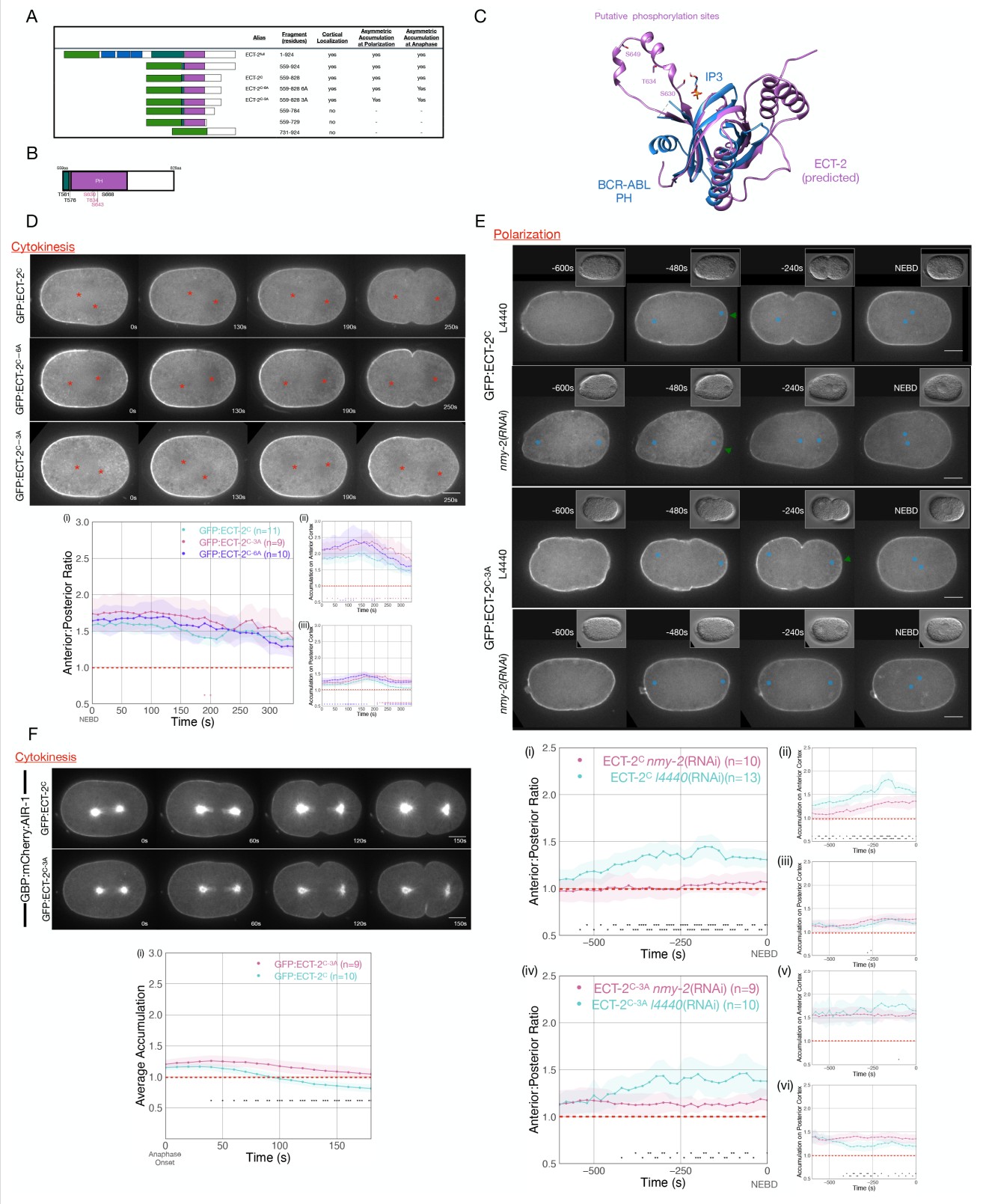

**Figure 5.** Phosphorylation of ECT-2 regulates its asymmetric cortical accumulation. (**A**) Schematic of GFP:ECT-2 truncations tested in this study and summary of their localization patterns. (**B**) Schematic of ECT-2<sup>C</sup> with locations of mutated residues. Residues in the loop in the PH domain are colored purple. (**C**) Alpha Fold prediction of the ECT-2 PH domain prediction (AF-Q9U364-F1) aligned with the crystal structure of the PH domain of Bcr-Abl (PDB 5OC7). (**D**) Stills from representative time series of GFP:ECT-2<sup>C</sup> either wild-type, 3A, or 6A mutants and the quantitation of accumulation of each

*Figure 5 continued on next page*

*Figure 5 continued*

transgene with NEBD as time 0. Source data is available (***Figure 5—source data 1***). (**E**) Asymmetric accumulation of GFP:ECT-2$^{C-3A}$ is lost when cortical flows are reduced. Images shown are from time-lapse acquisitions of either control or NMY-2-depleted embryos from transgenic lines expressing either GFP:ECT-2$^C$ or GFP:ECT-2$^{C-3A}$. Graphs (**i**) – (**iii**) quantify GFP:ECT-2$^C$ wild-type accumulation over time and graphs (**iv**)–(**vi**) quantify GFP:ECT-2$^{C-3A}$ accumulation over time, both with t=0 set as NEBD. Green arrowheads depict local reduction in ECT-2 as the centrosome is opposed to the posterior cortex. Source data is available (***Figure 5—source data 1***). (**F**) GFP:ECT-2$^{C-3A}$ is more refractory than GFP:ECT-2$^{C-WT}$ to GBP:mCherry:AIR-1 during cytokinesis. Panels show stills from time series of embryos co-expressing either GFP:ECT-2$^{C-WT}$ or GFP:ECT-2$^{C-3A}$ with GBP:mCherry:AIR-1. The graphs reflect GFP:ECT-2$^C$ cortical accumulation over time starting at anaphase onset. ECT-2 accumulation on the spindle was excluded from measurements. Source data is available (***Figure 5—source data 1***).

The online version of this article includes the following source data and figure supplement(s) for figure 5:

**Source data 1.** The source data is as described in legend to ***Figure 1—source data 1***.

**Figure supplement 1.** Endogenous ECT-2 was depleted in a GFP:ECT-2$^C$ expressing embryo.

**Figure supplement 2.** AIR-1 inhibits cortical accumulation of GFP:ECT-2$^C$.

**Figure supplement 2—source data 1.** The source data is as described in legend to ***Figure 1—source data 1***.

attenuated (***Figure 5E***). In 7/9 embryos, however, a local inhibitory effect in the immediate vicinity of the centrosome is detectable. In contrast, under these conditions, only 1/10 ECT-2$^{C-3A}$ embryos exhibit significant displacement near the centrosome. These results suggest that centrosome-associated AIR-1 locally inhibits cortical ECT-2 accumulation.

To test whether ECT-2 displacement from the cortex can be triggered by Aurora A, we co-expressed a derivative of AIR-1 tagged with the GFP-binding domain (GBP::mCherry::AIR-1) with fusions of GFP:ECT-2$^C$ and GFP:ECT-2$^{C-3A}$ (***Klinkert et al., 2019***). Upon anaphase onset, GBP::mCherry::AIR-1 induces displacement of GFP:ECT-2$^C$ (***Figure 5F***). Importantly, GFP:ECT-2$^{C-3A}$ is refractory to GBP::mCherry::AIR-1-induced displacement (***Figure 5F***). These results support the hypothesis that AIR-1 directly modulates ECT-2 localization by phosphorylation of residues within a loop in the PH domain.

## AIR-1 regulates centralspindlin-dependent furrowing

The AIR-1 pathway has been shown to impact centralspindlin-independent furrowing. For example, embryos deficient in centralspindlin subunits form ingressing, NOP-1-dependent furrows during cytokinesis which fail to form upon depletion of SAPS-1/PPH-6, a complex that negatively regulates AIR-1 (***Afshar et al., 2010***). Conversely, depletion of AIR-1 results in ectopic furrows during pseudocleavage (***Kapoor and Kotak, 2019***) and cytokinesis (***Figure 4A***). NOP-1, the upstream-most factor in the centralspindlin-independent pathway (***Tse et al., 2012***), is required for these ectopic furrows (***Kapoor and Kotak, 2019***).

To extend these findings, we first tested whether AIR-1 depletion bypasses the requirement for CYK-4 and NOP-1 in ECT-2 activation. To that end, we depleted both AIR-1 and CYK-4 in *nop-1(it142)* embryos. These triply deficient embryos fail to furrow during anaphase (100%, n=8). Next, we asked whether AIR-1 impacts centralspindlin-dependent furrowing, by depleting AIR-1 in NOP-1-deficient embryos. While 100% of embryos deficient in NOP-1 furrow to completion, ~30% embryos deficient in both AIR-1 and NOP-1 ingress with markedly slower kinetics and some fail to complete ingression (***Figure 6A***). These results suggest that while AIR-1 primarily impacts NOP-1-directed furrowing, it also affects centralspindlin-directed furrowing.Due to the role of AIR-1 in promoting nuclear envelope breakdown (***Hachet et al., 2007***; ***Portier et al., 2007***), embryos depleted of AIR-1 appear to initiate furrowing more rapidly when NEBD is used as a reference point.

Finally, we sought to confirm that the effects resulting from manipulating AIR-1 reflects the specific impact of this kinase on ECT-2. To that end, we generated a strain in which one of the putative phospho-acceptor residues in ECT-2, T634, was mutated to a glutamic acid. This strain could be readily maintained as a homozygote (>98% hatch rate), indicating that this allele retains significant function, although cortical accumulation of ECT-2$^{T634E}$ is reduced and less asymmetric. To test whether the T634E substitution impacts ECT-2 activity, we used RNAi to partially deplete ECT-2:mNG and ECT-2$^{T634E}$:mNG. Reduction of ECT-2:mNG levels to ~20% of wild-type results in cytokinesis failure in only ~11% of embryos. Thus, in the wild-type, ECT-2 accumulates to levels in excess of that required for completion of cytokinesis. In contrast, a similar reduction of ECT-2$^{T634E}$:mNG results in cytokinesis

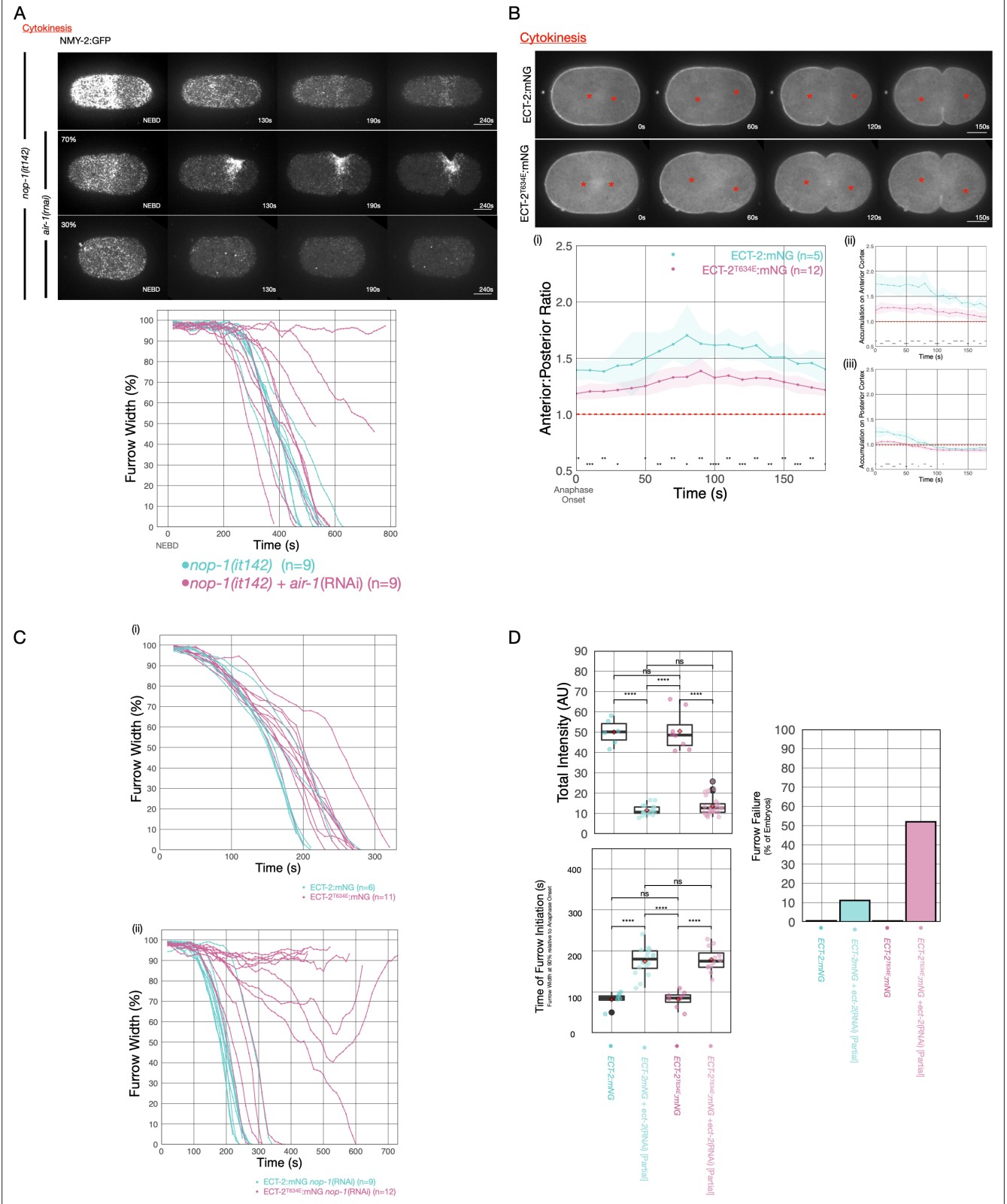

**Figure 6.** AIR-1 is involved in centralspindlin-dependent furrowing. (**A**) Furrowing is variable in *nop-1(it142); air-1*(RNAi) embryos. Shown are cortical maximum intensity projections 6 μm deep of NMY-2:GFP; *nop-1(it142)* embryos with or without depletion of AIR-1. Furrow width as a function of time starting at NEBD in individual embryos is shown graphically using a rolling average period of three time points. Note that NEBD is delayed in AIR-1-depleted embryos resulting in contractility at an earlier time point. Source data is available (*Figure 6—source data 1*). (**B**) ECT-2^T634E exhibits reduced

*Figure 6 continued on next page*

*Figure 6 continued*

cortical accumulation. Embryo images from time series of ECT-2:mNG or ECT-2$^{T634E}$:mNG expressing embryos in strains co-expressing mCherry:Tubulin. (i–iii) Graphs represent ECT-2 accumulation as a function of time with anaphase onset as time 0 s. Source data is available (*Figure 6—source data 1*). (**C**) ECT-2$^{T634E}$ slows furrow ingression and affects central spindle-dependent furrowing. (**i**) Furrow kinetics of ECT-2:mNG or ECT-2$^{T634E}$:mNG in strains co-expressing mCherry:Tubulin. Graphs represent the extent of furrow ingression of individual embryos as a function of time starting at anaphase onset using a rolling average of three time points. (ii) Furrow kinetics of ECT-2:mNG or ECT-2$^{T634E}$:mNG in strains co-expressing mCherry:Tubulin. Graphs represent the extent of furrow ingression of individual embryos as a function of time starting at anaphase onset using a rolling average of three time points. Source data is available (*Figure 6—source data 1*). (**F**) ECT-2$^{T634E}$:mNG is more dosage sensitive than ECT-2:mNG. (**i**) Average total intensity of ECT-2 in embryos measured and graphed as box whisker plots for control embryos or embryos partially depleted of ECT-2. Furrow kinetics of control or partially depleted ECT-2 embryos expressing ECT-2:mNG or ECT-2$^{T634E}$:mNG co-expressing NMY-2:mKate were measured and used to determine (ii) the time between anaphase onset and furrow initiation (ingression to 90% furrow width) and (iii) the fraction of embryos that failed division. All non-depleted embryos completed division. Source data is available (*Figure 6—source data 1*).

The online version of this article includes the following source data for figure 6:

**Source data 1.** This source data table contains (**I**) the measurements of furrow width at each time point of each embryo.

failure in ~50% of embryos, suggesting that phosphorylation of ECT-2 on T634 attenuates ECT-2 function. Importantly, whereas ECT-2$^{T634E}$:mNG slows the rate of furrow ingression as compared to wild-type, furrow initiation is not delayed. In contrast, partial depletion of ECT-2 does not affect the rate of furrow ingression, but it delays furrow initiation. These results suggest that the defects resulting from the T634E substitution are distinct from those resulting from an overall reduction in ECT-2 function. To test whether this substitution affects centralspindlin-dependent furrowing, we depleted NOP-1 in embryos expressing ECT-2:mNG and ECT-2$^{T634E}$:mNG. Whereas ECT-2:mNG embryos depleted of NOP-1 complete cytokinesis with only a slight delay, embryos expressing ECT-2$^{T634E}$ and depleted of NOP-1 exhibit highly variable furrowing behavior. While a minority of such embryos complete the first division (5/12), many embryos form furrows that ingress at variable rates and to variable extents (*Figure 6Bv*). We conclude that the introduction of a single phosphomimetic residue at a putative AIR-1 site in a flexible loop in the PH domain of ECT-2 impairs centralspindlin-dependent furrowing.

## Discussion

During cytokinesis, cleavage furrow formation is directed by a combination of a positive signal that spatially correlates with the position of the spindle midzone, and a negative signal associated with spindle poles. The positive cue is centralspindlin-dependent and is fairly well understood; far less is known about the inhibitory cue(s) associated with spindle poles. Here, we show that an inhibitory cue associated with centrosomes triggers a local, Aurora A kinase-dependent inhibition of cortical ECT-2 recruitment (*Figure 7A*). Our studies suggest that the posteriorly shifted spindle triggers preferential displacement of ECT-2 from the posterior cortex by Aurora A-dependent phosphorylation of the ECT-2 PH domain, though the evidence for this phosphorylation event is indirect. This local inhibition of cortical ECT-2 recruitment triggers anterior-directed, myosin-dependent cortical flows that amplify ECT-2 asymmetric accumulation, likely generating positive feedback (*Figure 7B*). Inhibition of cortical accumulation of ECT-2 by Aurora A influences both centralspindlin-independent and -dependent furrowing. These results provide insights into the mechanism of generation of cortical asymmetries during polarization and cytokinesis and extends the commonalities between the two processes.

### Centrosomal AIR-1 regulates cortical association of ECT-2

In the early *C. elegans* embryo, shortly after fertilization, the RHO-1 GEF ECT-2 is enriched on the cortex relative to the cytoplasm. Following an initial period of uniform cortical association, as the paternally contributed centrosome approaches the cortex, ECT-2 accumulation at the nascent posterior cortex is reduced, coupled with an accompanying increase of ECT-2 in the nascent anterior cortex leading to the asymmetric distribution of ECT-2 and establishment of the primary body axis (*Motegi and Sugimoto, 2006*). The centrosome-induced reduction of posterior ECT-2 involves AIR-1 (*Zhao et al., 2019*; *Kapoor and Kotak, 2019*; *Klinkert et al., 2019*).

A C-terminal fragment of ECT-2, lacking the regulatory BRCT domains and the catalytic GEF domain, largely recapitulates the asymmetric distribution of endogenous ECT-2 throughout the early divisions. The PH domain contained in this fragment contains a sequence, predicted to form a loop,

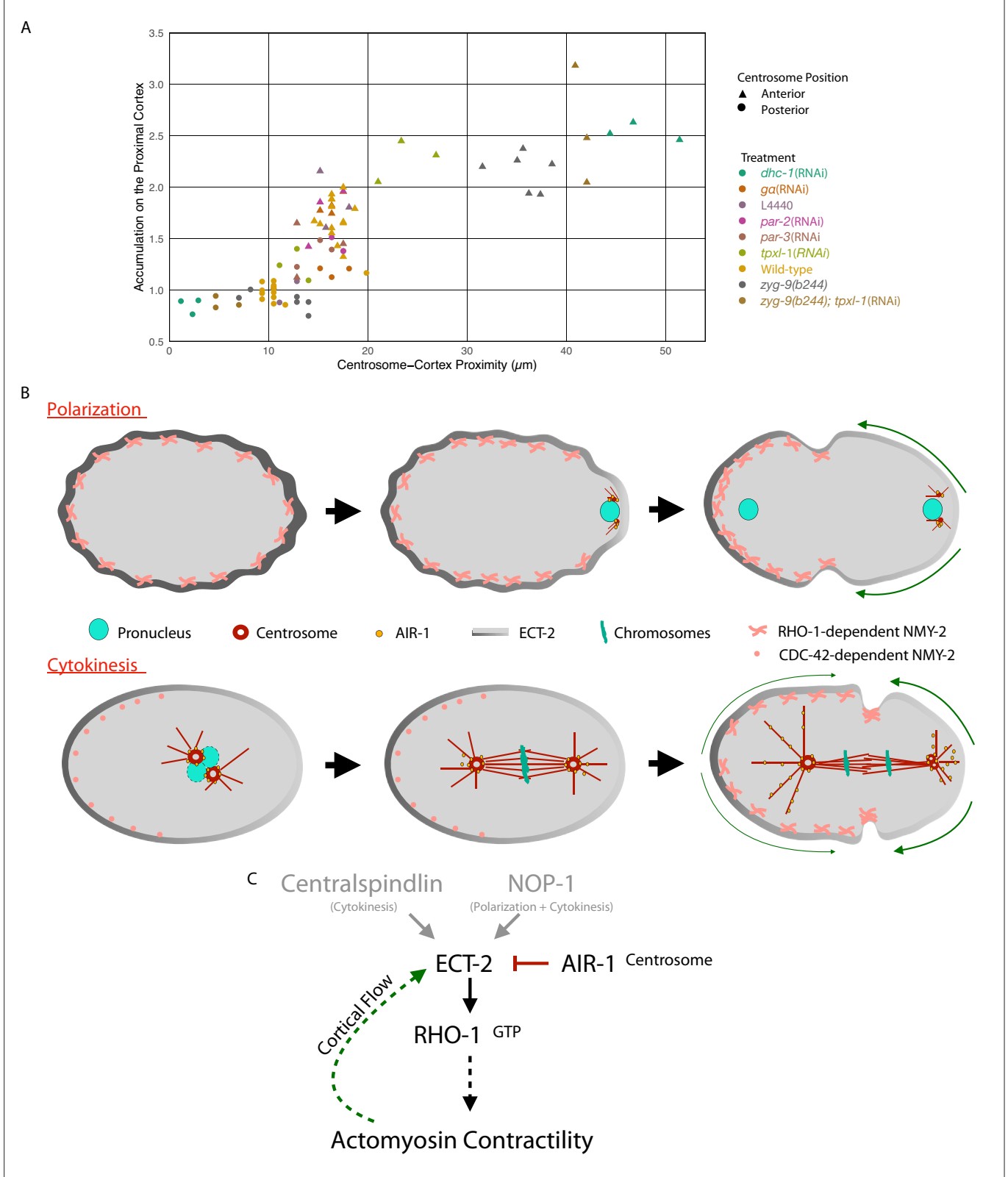

**Figure 7.** Model of local symmetry breaking of cortical ECT-2 and amplification by cortical flows. (**A**) Summary graph of cortical ECT-2 accumulation at the time of maximal ECT-2 asymmetry as a function of distance between the centrosome and the proximal cortex. Centrosome-cortex distance was measured using either a mCherry:Tubulin or Nomarski image. Three embryos from each treatment listed in the legend were chosen at random. Source data is available (**Figure 7—source data 1**). (**B**) During both polarization and cytokinesis, centrosome-derived AIR-1 locally inhibits cortical ECT-2

*Figure 7 continued on next page*

Figure 7 continued

accumulation, triggering symmetry breaking and generation of cortical flows directed away from the site of symmetry breaking. ECT-2 dynamically associates with components that undergo cortical flow, leading to its further depletion from the posterior cortex and its anterior accumulation. During polarity establishment, the centrosomes are small and closely apposed to the cortex, whereas during anaphase the centrosomes are large and accumulate significant AIR-1 that modulate cortical association of ECT-2. During polarization, the pseudocleavage furrow forms at the boundary between low and high ECT-2 and myosin levels; in cytokinesis, this site correlates with the centralspindlin-dependent furrow that correlates with the spindle midzone. (**C**) Proposed genetic pathway (**D**).

The online version of this article includes the following source data for figure 7:

**Source data 1.** This data lists the distance of each anterior or posterior centrosome to its respective membrane (anterior or posterior) and the amount of ECT-2 accumulation on the respective membrane.

that contains several putative sites for phosphorylation by AIR-1. Mutation of these sites to non-phosphorylatable alanine residues enhances ECT-2 cortical association and renders it resistant to the inhibitory action of AIR-1; conversely, phosphomimetic mutations reduce its cortical association. We hypothesize that AIR-1 phosphorylates these sites thereby inhibiting ECT-2 cortical association. The flexible loop containing these putative sites is predicted to lie adjacent to the surface where phosphoinositides bind in structurally characterized PH domains (*Reckel et al., 2017*). The strong effect of phosphomimetic and non-phosphorylatable mutations on ECT-2 accumulation is striking, given the predicted flexibility of this loop, lending additional support to their relevance.

Our results implicating AIR-1 in polarity establishment are largely in concordance with those of *Zhao et al., 2019*, with the exception that these authors did not observe an effect of AIR-1 depletion on ECT-2 localization during polarity establishment. In that regard, our results align with those of *Kapoor and Kotak, 2019*. Multiple lines of evidence implicate AIR-1 in inhibiting ECT-2 cortical localization. These include (i) changes in cortical accumulation of ECT-2 upon depletion of SAPS-1 and AIR-1 (ECT-2$^C$ is also enhanced upon AIR-1 depletion); (ii) increased cortical accumulation of GFP:ECT-2$^{C-3A}$ as compared to GFP:ECT-2$^C$; (iii) susceptibility of GFP:ECT-2$^C$, but not GFP:ECT-2$^{C-3A}$ to GBP:mCherry:AIR-1; (iv) a phosphomimetic substitution of a predicted AIR-1 site in ECT-2, T634, attenuates its cortical accumulation and function.

During both polarization and cytokinesis, centrosomes, by virtue of their role in concentrating and activating AIR-1 (*Hamill et al., 2002*; *Joukov et al., 2010*), appear to act as cues for ECT-2 displacement. This assertion is based on several earlier studies (*Kapoor and Kotak, 2019*; *Klinkert et al., 2019*) and the observations shown here. Specifically, during both processes, ECT-2 displacement requires the core centrosomal component SPD-5, which is required to recruit AIR-1 to centrosomes (*Hamill et al., 2002*), but ECT-2 displacement is not inhibited by depolymerization of microtubules and it does not require the AIR-1 activator TPXL-1 (see below). During cytokinesis, centrosomes preferentially inhibit ECT-2 cortical association in the posterior of the embryo. This difference results from the asymmetric position of the spindle that arises as a consequence of embryo polarization.

To summarize our results related to cytokinesis, we plotted cortical ECT-2 accumulation as a function of centrosome-cortical distance in representative embryos from a variety of experimental conditions (*Figure 7A*). These data demonstrate that, during cytokinesis, ECT-2 accumulation and centrosome proximity inversely correlate over a range of distances up to ~20 μm (40% of embryo length). The inhibitory effect of centrosomes on ECT-2 accumulation decays rapidly in the range of 10–18 μm from the cortex, which correspond to the positions of the posterior and anterior centrosomes in wild-type embryos (*Kemphues et al., 1988*), respectively. These findings are consistent with our earlier work demonstrating that spindle elongation regulates furrowing during cytokinesis, although we previously inferred that the inhibition was due to astral microtubules, which are proximal to the cortex (*Dechant and Glotzer, 2003*), rather than centrosomes which are not.

## Cortical flows amplify ECT-2 asymmetry

Centrosomes and AIR-1 kinase promote ECT-2 asymmetry during both polarity establishment and cytokinesis. During polarity establishment in myosin-depleted embryos which lack cortical flows, the zone of inhibition of ECT-2 accumulation in the posterior shrinks considerably; similar observations have reported previously (*Motegi and Sugimoto, 2006*). During cytokinesis, perturbations, such as microtubule disassembly and ZYG-9 depletion, enhance cortical flows creating a larger posterior zone of inhibition and enhance the accumulation of ECT-2 in the anterior. The enhancement of these flows

is likely the result of a combination of a reduction in the distance between the posterior centrosome to the cortex and the removal of a centrosome in the anterior domain which would otherwise suppress contractility in that region. The ability of ECT-2$^{C-3A}$ to polarize in the presence, but not the absence of myosin, further establishes the role of flows in amplifying ECT-2 asymmetry. These results suggest that in wild-type embryos, Aurora A-dependent symmetry breaking is amplified by anterior-directed cortical flows that enhance ECT-2 asymmetry. Further, we speculate that centrosomal AIR-1 not only breaks symmetry, but that this regulation of ECT-2 by centrosomal AIR-1 continues throughout anaphase. These findings extend our understanding of the mechanistic basis for the requirement for myosin in polarity establishment (*Guo and Kemphues, 1996*). Furthermore, asymmetric ECT-2 is predicted to enhance asymmetric assembly of cortical myosin, which is predicted to accentuate cortical flows, further amplifying ECT-2 asymmetry.

Long-range flows result from the viscosity of the cortical actomyosin network (*Mayer et al., 2010*). It is unlikely that ECT-2 is concentrated by these flows itself, as it exchanges rapidly, on the order of seconds. Rather, we favor a model in which the association of ECT-2 with the cortex involves inter-actions with cortical component(s) that are concentrated by cortical flows (*Mayer et al., 2010*). This would explain the anterior accumulation of ECT-2 as polarization proceeds and the enhanced ECT-2 asymmetry during cytokinesis in embryos with exaggerated flows. The PH domain of ECT-2 has been shown to bind the actomyosin-binding protein anillin (*Frenette et al., 2012*), making it a candidate contributor to ECT-2 cortical anchoring. Anterior PAR proteins may also contribute to ECT-2 accumu-lation. Interestingly, anterior PAR proteins are known to promote cortical flow (*Munro et al., 2004*). Further analysis of the interactions required for ECT-2 association will be a focus of future studies.

Although RhoGEF activators accelerate cortical flows, basal levels of myosin appear sufficient to drive the cortical flows that polarize ECT-2. For example, inactivation of NOP-1 slows cortical flows and delays the polarization of anterior PAR proteins, but it does not preclude asymmetric accumula-tion of ECT-2 (*Rose et al., 1995*; *Tse et al., 2012*) and this study. Cortical flow-mediated concentration of ECT-2 may be central to polarization of the embryo, analogous to flow-induced re-orientation of PAR protein polarity (*Mittasch et al., 2018*).

During anaphase, asymmetric ECT-2 accumulation is also myosin-dependent, presumably due to its role in generating cortical flows. During cytokinesis, basal myosin levels appear to be sufficient to promote asymmetric ECT-2 accumulation, as ECT-2 asymmetry increases during anaphase even when both NOP-1 and CYK-4 are attenuated. We infer that bulk, cortical ECT-2 has a low level of RhoGEF activity.

In summary, during both polarization and cytokinesis, the cortex undergoes large, furrow-inducing, flows of contractile cortical material due to the influence of the centrosomes on ECT-2 localization. During polarization the two centrosomes are tiny and they interact with the male pronucleus (*Malone et al., 2003*). These immature centrosomes are usually closely associated with the posterior cortex (*Delattre et al., 2006*; *Pelletier et al., 2006*; *Cowan and Hyman, 2004*). By contrast, during the first cytokinesis, the two centrosomes are large, they contain dramatically more AIR-1 kinase (*Hannak et al., 2001*), and they are separated from each other by the length of the spindle and they lie at least 10 μm from the nearest cortex (*Figure 7A*). Despite the wide differences in size, components, and positions of the immature and mature centrosomes, roughly similar patterns of ECT-2 accumulation and contractility are induced during polarization and cytokinesis.

## Cell cycle regulation of ECT-2 cortical localization

The temporal pattern of ECT-2 cortical accumulation is dynamic. ECT-2 is initially uniformly distributed on the cortex and becomes progressively asymmetric through the first interphase. Cortical ECT-2 accumulation then declines throughout maintenance phase, though asymmetry remains. During cyto-kinesis, ECT-2 levels on the cortex decline, though it is preferentially displaced from the posterior cortex, coincident with spindle elongation.

While many of these changes in cortical localization reflect changes controlled by AIR-1 and centro-some position, other factors appear to influence cortical accumulation of ECT-2. For example, cell cycle regulated, global changes in ECT-2 accumulation are apparent in AIR-1-depleted embryos. Further-more, the ability of GBP::mCherry::AIR-1 to induce GFP-ECT-2C displacement fluctuates with the cell cycle, suggesting the susceptibility of ECT-2 to AIR-1 may be cell cycle regulated. For example, Cdk1 phosphorylation of ECT-2 might inhibit its phosphorylation by AIR-1.

## TPXL-1 is not required for ECT-2 asymmetry

In addition to examining the role of AIR-1 in regulating cortical association of ECT-2, we also examined the role of one of its activators, TPXL-1 (*Ozlü et al., 2005*). TPXL-1 targets AIR-1 to astral microtubules, which are well positioned to modulate cortical ECT-2. However, TPXL-1 makes a more modest contribution to ECT-2 asymmetry than does AIR-1. Additionally, the effect observed upon TPXL-1 depletion might be, at least in part, an indirect consequence of the role of TPXL-1 in regulating spindle assembly and positioning. Indeed, TPXL-1 is dispensable for ECT-2 asymmetry when spindle poles are close to the cortex. This finding is consistent with earlier work that revealed that several AIR-1 functions, particularly those at the centrosome, are TPXL-1 independent (*Ozlü et al., 2005*; *Zhao et al., 2019*).

A previous study demonstrated that TPXL-1 is involved in polar relaxation during cytokinesis (*Mangal et al., 2018*). There are several differences between that study and the present one. Here, we assayed ECT-2 accumulation, whereas the previous study assayed accumulation of the RHO-1 effector anillin in embryos that were depleted of both myosin II and the principle RhoGAPs that inhibit RHO-1, RGA-3/4. In that context, TPXL-1 appears to inhibit accumulation of anillin specifically at the anterior cortex; anillin does not strongly accumulate on the posterior cortex of either wild-type or TPXL-1-depleted embryos (*Mangal et al., 2018*). Thus, AIR-1 appears to regulate Rho-mediated contractility at multiple steps, which are differentially dependent on both TPXL-1 and NMY-2.

## Centralspindlin can facilitate ECT-2 cortical association

Despite the inhibition of ECT-2 accumulation at the posterior cortex, which is further potentiated when the spindle is displaced to the posterior, a furrow can form at a site adjacent to the spindle midzone in the embryo posterior. Formation of this furrow requires centralspindlin (*Werner et al., 2007*). Despite the overall inhibition of ECT-2 cortical association in the posterior, our results suggest an initial centralspindlin accumulation recruits and activates ECT-2 to induce furrowing at this site.

These results indicate that cortical recruitment of ECT-2 can occur by two distinct modes. One mode is independent of centralspindlin, and is inhibited by AIR-1. The other mode is dependent on centralspindlin and is less sensitive to AIR-1 activity. The centralspindlin-dependent mode of ECT-2 accumulation correlates with a reduced mobile fraction of ECT-2 as compared to the ECT-2 that accumulates on the cortex more broadly. Centralspindlin-dependent accumulation of cortical ECT-2 is reminiscent of its regulation in cultured human cells. In HeLa cells, Ect2 does not constitutively associate with the cortex, due to Cdk1-mediated phosphorylation of a basic region C-terminal to the PH domain (*Su et al., 2011*). During anaphase in these cells, equatorial accumulation of ECT-2 requires centralspindlin (*Su et al., 2011*). Thus, in both systems, phosphorylation of the C-terminus of ECT-2 attenuates its cortical association. However, the kinases are distinct (Cdk1 in human cells vs. AIR-1 in *C. elegans*) whereas the phosphoregulation appears global in human cells and spatially regulated in *C. elegans* blastomeres.

## Polar relaxation in *C. elegans* and mammalian cells

The results shown here indicate that inhibition of ECT-2 accumulation by centrosomal Aurora A at spindle poles can induce furrow formation. The conservation of centrosomal Aurora A and ECT-2 suggests that this phenomenon could be generalizable. In mammalian cells, Aurora A may function in parallel to Cdk1 (*Su et al., 2011*) to antagonize cortical association of Ect2 at certain times of the cell cycle. A recent report, relying primarily on chemical inhibitors of Aurora kinases, indicates that Aurora B promotes clearance of F-actin from the polar cortex (*Ramkumar et al., 2021*). However, these inhibitors are not highly selective for Aurora A vs. Aurora B at the concentrations used in cell-based assays (*de Groot et al., 2015*).

Remarkably, the highly similar kinases, Aurora A and Aurora B, regulate cytokinesis in quite different manners. Aurora A primarily acts to inhibit contractility at cell poles through ECT-2, while Aurora B primarily promotes contractility at the equator through centralspindlin (*Basant et al., 2015*).

## Limitations of this study

While our data implicate AIR-1 in phosphorylation of ECT-2, we have not directly shown that AIR-1 phosphorylates ECT-2. Additionally, while mutations of these sites affect ECT-2 localization, we have not shown that these sites are phosphorylated in vivo.

## Materials and methods

### Plasmid and strain construction

#### MosSCI transgenic strains

*C. elegans* strains expressing GFP:ECT-2$^C$ and GFP:ECT-2$^{C-3A}$, and GFP:ECT-2$^{C-6A}$ using the promoter and 3'utr elements from *pie-1*. These transgenes were integrated into ttTi5605 on Chromosome II using MosSCI (*Frøkjær-Jensen et al., 2012*; *Frøkjaer-Jensen et al., 2008*). A complete list of strains used in this study is available (*Supplementary file 1*). Plasmids were built using Gibson Assembly (*Gibson et al., 2009*). A complete list of plasmids used in this study is available (*Supplementary file 2*). A complete list of oligonucleotides used in this study is available (*Supplementary file 3*).

#### CRISPR

mNG was integrated at the C-terminus of endogenous ECT-2 using CRISPR (*Dickinson et al., 2015*). ECT-2$^{T634E}$ was mutated at the endogenous locus by CRISPR using Cas9 ribonucleoprotein [IDT] complexes (crRNA sequence: ACGCTGTCTCGAATGTAAAC) and a single-stranded oligodeoxynucleotide (CAGCGAGAAGAATgaAATGCAACGTCTAGCTCGTCAtGCGTCGTTTGCgAGTTTACATTC) (*Paix et al., 2015*; *Dokshin et al., 2018*).

#### RNAi vector construction

ECT-2(244-536aa)/L4440 and TPXL-1(50-395aa)/L4440 were amplified from N2 lysates and inserted into L4440 between SacI and SpeI by Gibson Assembly (*Gibson et al., 2009*). For TBA-2 + TBB-2/L4440, the TBB-2 fragment from TBB-2/L4440 (Ahringer library) was amplified and inserted into the KpnI site of TBA-2/L4440 (Ahringer library) by ligation.

### RNAi

Ahringer library: *par-3, air-1, spd-5, saps-1, dhc-1, perm-1, nop-1*, and *zyg-9* (*Kamath et al., 2003*).

*ect-2* [this study]*, tpxl-1* [this study]*, tba-2+tbb-2* [this study]*, cyk-4* (*Zhang and Glotzer, 2015*), *gα* (from Pierre Gönczy), and *nmy-2* (from Ed Munro).

For all experiments, RNAi plasmids were transformed into HT115 competent cells and cultured for 6–8 hr in LB + ampicillin at 37°C; cultures or mixtures thereof were seeded onto NGM plates containing ampicillin and 1 mM IPTG and incubated overnight at room temperature before storage at 4° C.

For *ect-2:* RNAi plates were seeded with a mixed ECT-2:L4440 culture (1:2 or 1:10). For experiments, L4 worms were transferred to plates about 20 hr prior to imaging.

For *perm-1*: RNAi plates were seeded with a mixed PERM-1:L4440 culture (1:20) (*Carvalho et al., 2011*). For experiments, L4 worms were transferred to plates about 16 hr prior to imaging.

For all experiments except as indicated, L4 hermaphrodites were transferred to RNAi plates for ~24 hr before imaging.

RNAi-treated embryos were phenotypically assessed for sufficient depletion. These phenotypes were assessed using Nomarski or mCherry:Tubulin. Embryos that did not show these phenotypes were excluded from the analysis:

> *air-1*(RNAi): Mitotic spindle fails to assemble properly; hypercontractility during polarization (*Hannak et al., 2001*).
>
> *cyk-4*(RNAi): Loss of bundling of anti-parallel microtubules upon anaphase onset, exaggerated anaphase B (*Jantsch-Plunger et al., 2000*).
>
> *dhc-1*(RNAi): Lack of pronuclear migration and centrosomes remained in posterior during polarization and cytokinesis (*Gönczy et al., 1999*).
>
> *ect-2*(RNAi) (partial): Lack of cortical ruffling during polarization (*Motegi and Sugimoto, 2006*).
>
> *gα*(RNAi): Absence of spindle rocking during anaphase and shorter aster separation (*Gotta and Ahringer, 2001*).
>
> *nmy-2*(RNAi) (partial): Lack of pseudocleavage and contractility during polarization (*Munro et al., 2004*).
>
> *nop-1*(RNAi): Lack of pseudocleavage (*Rose et al., 1995*; *Tse et al., 2012*).
>
> *par-2*(RNAi): Symmetric first division (*Kemphues et al., 1988*).
>
> *par-3*(RNAi): Symmetric first division (*Etemad-Moghadam et al., 1995*).

saps-1(RNAi): Absence of spindle rocking during anaphase (*Afshar et al., 2010*).
spd-5(RNAi): Lack of pronuclear migration and spindle assembly failure (*Hamill et al., 2002*).
tba-2 + tbb-2(RNAi) + nocodazole: Centrosomes remained in posterior during polarization and cytokinesis and asters did not separate during cytokinesis (*Tsai and Ahringer, 2007*).
tpxl-1(RNAi): Small mitotic spindle (*Ozlü et al., 2005*).
zyg-9(RNAi): Spindle assembles in posterior of the embryo (*Matthews et al., 1998*).

## Imaging

### Mounting

Generally, gravid *C. elegans* were dissected in egg salts (5 mM HEPES pH 7.4, 118 mM NaCl, 40 mM KCl, 3.4 mM MgCl$_2$, 3.4 mM CaCl$_2$) and mounted on 2% agarose pads. For tubulin experiments, *C. elegans* were dissected in egg salts plus 50 µg/mL nocodazole or DMSO.

For MLN8237 inhibitor experiment: After *perm-1* depletion, *C. elegans* were dissected in embryonic imaging medium (50% L-15 Leibovitz's Medium [Gibco], 10% 5 mg/mL Inulin, 20% FBS [Gibco], 20% 250 mM HEPES pH 7.5) on a Poly-D-Lysine (R&D Systems)-coated coverslip mounted on a slide with double-sided tape (3M) used as spacer. Additional imaging media was flowed into fill the chamber. Fifty µM MLN8237 or DMSO was flowed into the embryonic chamber at 100 s post NEB while imaging.

For imaging early polarization: Worms were dissected into embryonic imaging medium (50% L-15 Leibovitz's Medium [Gibco], 10% 5 mg/mL Inulin, 20% FBS, 20% 250 mM HEPES, pH 7.5) on a Poly-D-Lysine (R&D Systems)-coated coverslip mounted on a slide with double-sided tape (3M) used as spacer. Additional media was flowed into fill the chamber before imaging.

### Microscopy

Time-lapse recordings of embryos were acquired with a Zeiss Axioimager M1 equipped with a Yokogawa CSU-X1 spinning disk unit (Solamere) using 50 mW 488 and 561 nm lasers (Coherent) and a Prime 95B camera (Photometrics). Images were taken at 63× using Micromanager software. ECT-2:mNG was imaged at 10 s intervals during cytokinesis and at 20 s intervals during polarization. NMY-2:GFP was imaged at 10 s intervals during polarization and cytokinesis. In all cases, a Nomarski image was acquired at each time point. Nuclear envelope breakdown was scored based on ECT-2 fluorescence and/or the appearance of the nucleus in the Nomarski images. For the NMY-2:GFP particle tracking, GFP was imaged only at the cortex by taking 4-1 µm step slices at a stream rate beginning at anaphase onset.

FRAP experiments were performed on a Zeiss LSM 880 equipped with AiryScan on confocal setting using a Plan-Apochromat 63×/1.40 Oil DIC M27 objective. Images were taken at 100 ms intervals.

## Image analysis

All embryos were processed and analyzed using FIJI (*Schindelin et al., 2012*). Further analysis and graphs were generated using RStudio with the following packages: tidyverse, gcookbook, ggplot2, broom, ggpubr, rlang, mgcv, and zoo (*Wickham et al., 2019*).

### ECT-2 accumulation

For each embryo, at every time point, the average intensities of the background and cytoplasm were measured. The embryonic cortex was identified by performing a Gaussian blur on a copy of the embryo, using the FIJI magic wand to select the embryo interior. This area was converted to a line, transferred to the original image and then the line was expanded to a width of 50 pixels and straightened from the middle of the anterior of the embryo clockwise around the perimeter (*Figure 1—figure supplement 1*). From the image containing the straightened cortical region, a 3 pixel-wide line was drawn perpendicular to the cortex and the maximum intensity was recorded at each position along the line. Accumulation was calculated for every position using (maximum intensity-background)/(cytoplasm-background). The measurements of anterior and posterior accumulation reflect the average accumulation in the anterior 20% or the posterior 20% of the embryo, respectively.

The perimeter was measured as the length of the ROI used to identify the embryo cortex at each time point. The perimeter was normalized to the maximum perimeter of each embryo. To assess

boundary length, ECT-2:mNG accumulation across all positions along the perimeter of the embryo was fitted to a regression model by generalized additive model (GAM) (prediction accumulation) in R Studio using the mgcv:gam function. The average cortical accumulation over the anterior 60% of the embryo was calculated and a threshold was set at 85% of this average. The number of positions that fell below the threshold were counted and the boundary length was set as the fraction of positions below threshold.

The accumulation profile of ECT-2 was calculated from the average of the regression models at the time during cytokinesis at which each embryo exhibited maximally asymmetric ECT-2. This estimate is plotted as a function of relative position along the cortex of the embryo with 0 as the anterior and 0.5 as the posterior.

For the ect-2(RNAi) partial depletion experiments, embryo intensity was measured as the average pixel intensity of the embryo after subtracting the background at NEBD.

### Furrow kinetics

The positions of the furrow tips in NMY-2:GFP, NMY-2:mKate, or ECT-2:mNG embryos were measured at 10 s intervals starting at NEBD or anaphase. The distance between the furrow tips were normalized to maximum embryo width [furrow width]. Furrow width was plotted as a rolling average (period = 3) over time for individual embryos.

For the partial ect-2(RNAi) experiments: To measure ingression initiation, the length of time between anaphase onset and 90% width was measured and plotted. For furrow failure, the fraction of embryos that did not reach a furrow width of 0% were counted and plotted.

### FRAP

The FRAP images were analyzed using FIJI. The FRAP Analysis Script source is https://imagej.net/imagej-wiki-static/Analyze_FRAP_movies_with_a_Jython_script.

The coefficients from the fitted recovery curves of each individual embryo were averaged for each condition and the average curve was plotted.

### Centrosome-cortical distance

Three embryos of each treatment indicated in the legend were chosen at random and, the time point of maximally asymmetric ECT-2 accumulation was selected for measurement. Centrosome distance to the nearest cortex was measured using the mCherry:Tubulin or Nomarski images. This distance and the average ECT-2 accumulation at the nearest cortex were plotted.

### Particle tracking

Myosin particles were tracked using the Trackmate plugin in FIJI (*Ershov et al., 2022*).

### Statistical tests

Ninety-five percent confidence intervals are shown as shaded ribbons in the ECT-2 cortical accumulation graphs. Additionally, t-tests were performed in R Studio to test the significance of ECT-2 accumulation between two treatments (i.e. RNAi depletion and wild-type) at every time point imaged as indicated in the figure legends. t-Tests were also performed to test the significance of the length of ECT-2 boundaries. Labels on the graphs are defined as follows: 'ns', $p>0.05$; *$p<0.05$; **$p<0.01$; ***$p<0.001$; ****$p<0.0001$.

## Structural analysis

The MatchMaker function of UCSF Chimera (*Pettersen et al., 2004*) (https://www.cgl.ucsf.edu/chimera/) was used to align the predicted PH domain of C.e. ECT-2 (https://alphafold.com/entry/Q9U364) to 5OC7 (*Reckel et al., 2017*).

## Materials availability

Strains (*Supplementary file 1*) and plasmids (*Supplementary file 2*) generated in this study are available from the author upon request. Images, analysis scripts, image quantification data, and plasmid maps will be deposited in Zenodo and associated with the DOI:10.7554/eLife.83992.

## Acknowledgements

We thank Dan Dickinson (UT, Austin) for sharing the LP229 strain and Pierre Gönczy (EPFL) for sharing the strain expressing GBP::AIR-1. We also thank Pierre Gönczy, Ed Munro (U of Chicago), and Ashley Rich (Duke University) for helpful comments on the manuscript. Some strains were provided by the CGC, which is funded by NIH Office of Research Infrastructure Programs (P40 OD010440). This work was supported by NIH grant R35GM127091.

## Additional information

### Funding

| Funder | Grant reference number | Author |
|---|---|---|
| National Institute of General Medical Sciences | R35GM127091 | Katrina M Longhini |

The funders had no role in study design, data collection and interpretation, or the decision to submit the work for publication.

### Author contributions

Katrina M Longhini, Data curation, Formal analysis, Investigation, Methodology, Writing – original draft, Writing – review and editing; Michael Glotzer, Conceptualization, Data curation, Formal analysis, Supervision, Funding acquisition, Methodology, Writing – original draft, Project administration, Writing – review and editing

### Author ORCIDs

Katrina M Longhini  http://orcid.org/0000-0002-6600-7083
Michael Glotzer  http://orcid.org/0000-0002-8723-7232

### Decision letter and Author response

Decision letter https://doi.org/10.7554/eLife.83992.sa1
Author response https://doi.org/10.7554/eLife.83992.sa2

## Additional files

### Supplementary files

- Supplementary file 1. List of *Caenorhabditis elegans* strains used in this study.
- Supplementary file 2. List of plasmids used in this study.
- Supplementary file 3. List of oligonucleotides used in this study.
- MDAR checklist

### Data availability

The data generated or analyzed during this study are included in the manuscript and in the associated source data files. Image files, raw image analysis data, image analysis scripts, R data files and plasmid sequences are uploaded to Zenodo at https://doi.org/10.5281/zenodo.7415982.

The following dataset was generated:

| Author(s) | Year | Dataset title | Dataset URL | Database and Identifier |
|---|---|---|---|---|
| Longhini KM, Glotzer M | 2022 | Aurora A and cortical flows promote polarization and cytokinesis by inducing asymmetric ECT-2 accumulation - Data Archive | https://doi.org/10.5281/zenodo.7415982 | Zenodo, 10.5281/zenodo.7415982 |

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
