## [Editor Report]

Cytokinesis in animals involves a contractile actomyosin ring, which generates the forces needed for cell division. A key factor controlling actomyosin ring function is a protein called ECT2, which is a regulator of the signalling protein RhoA GTPase. How ECT2 gets to the correct cellular location and how it executes its functions remain a mystery, despite extensive work. This work through a set of beautiful and thorough experiments establishes the mechanism of ECT2 intracellular distribution, which involves integration of spatial and biochemical signals all contributing to the fidelity of cell division.

---

## [Decision Letter]

[Editors' note: this paper was reviewed by Review Commons.]

Thank you for submitting your article "Aurora A and cortical flows promote polarization and cytokinesis by inducing asymmetric ECT-2 accumulation" for consideration by *eLife*. Your article has been reviewed by 3 peer reviewers at Review Commons and an *eLife* referee, and the evaluation at *eLife* has been overseen by a Reviewing Editor (Mohan Balasubramanian) and Anna Akhmanova as the Senior Editor.

Based on the previous reviews and the revisions, the manuscript has been improved but there are some remaining issues that need to be addressed, as outlined below:

The *eLife* expert raised some points which are transmitted verbatim below. I have read them carefully. I think the vast majority of points can be addressed by rewriting and providing explanations or toning down some of the conclusions. Also, the referee has asked that you provide some more experimental details, statistical methods, and additional citations.

Points arising from your response letter and revision:

1. In a revised manuscript, I am not convinced with their interpretation of the phenotype of air-1(RNAi);par-2(RNAi) zygotes. Given that either single par-2(RNAi) or single air-1(RNAi) abolished the anterior-enriched distribution of ECT-2::mNG (Figure 4 supplement 2), this data simply indicates the indispensable roles of both PAR-2 and AIR-1 in ECT-2 asymmetry, but they cannot conclude that ECT-2 asymmetry in air-1(RNAi) condition is due to PAR-2-dependent weaker cortical flows. Indeed, the anterior/the posterior ratio of ECT-2::mNG in air-1(RNAi) zygotes shown in Figure 4 supplement 2 is very close to 1.0 throughout mitosis, which is quite different from that in Figure 4A. This discrepancy should be addressed in the final manuscript.

2. The authors' response "While cytokinesis generally involves an equatorial contractile ring, furrow formation can be driven by an asymmetric – i.e. non-equatorial – accumulation of actomyosin. This behavior is exemplified during pseudocleavage during which the entire anterior cortex is enriched for actomyosin and the posterior is depleted of myosin (Figure 1 Supplement 2). Several published studies provide evidence that the asymmetric pattern of myosin accumulation contributes to cytokinesis (PMID 22918944, 17669650)."

The role of switching off the cortical flow from the P-to-A alone mode to the bidirectional mode in cytokinetic furrow formation has been reported in many papers (PMID: 27719759, 29963981, 32497213, etc.) as mentioned above. A simple unidirectional asymmetry is not sufficient in discussing the spatial regulation of cytokinesis.

3. The authors' response "However, as seen in e.g. ZYG-9 depleted embryos, ECT-2 is recruited to the posterior cortex in a centralspindlin-dependent manner". I don't understand the logic here. Has this been directly tested by, for example, depletion of ZEN-4 or CYK-4 in zyg-9(b244) mutant embryos? In Figure 3C (zyg-9(b244)), no particular enrichment of ECT-2 was observed at the posterior furrow, which is formed by centralspindlin.

4. "In contrast, the Gomez-Cavasos paper (PMID 32619481) shows in figure S2 that the PH domain is required for cortical localization of ECT-2; this paper does not focus extensively on the cortical accumulation of ECT-2". I think Gomez-Cavasos should also be cited as these provide complementary information as to the role of the PH domain.

5. In addition to the above, please answer either by rewriting or with experiments the points raised by the *eLife* referee below. As I read it, some experiments on Ect2 localization are required to firmly test your models. All other points raised might involve a balanced discussion of various observations and raising the limitations of the work.

*eLife* referee comments verbatim:Asymmetric actomyosin contractility plays key roles in various cellular activities. In *C. elegans* embryos, both the post-fertilization polarity establishment and cytokinesis depend on this process, which is known to be under the control of Rho GTPase. In this manuscript, the authors studied the molecular mechanism of symmetry breaking, focusing on ECT-2 RhoGEF, the crucial upstream activator of Rho, and its regulation by AIR-1, Aurora A kinase. Although the roles of these molecules both in polarity establishment and cytokinesis have already been reported, it remains unclear whether and how AIR-1 might regulate the activity of ECT-2.

By a combination of genetic manipulation and high-quality quantitive microscopy, the authors compared various perturbations and concluded that centrosomes and the cortical flow driven by actomyosin network play roles in the asymmetric cortical localization of ECT-2 while astral microtubules and TPXL-1, a conserved Aurora A regulator that recruits AIR-1 to the astral microtubules, are not essential for the ECT-2 asymmetry during cytokinesis in contrary to the previous reports. Then, the authors tested the hypothesis that AIR-1 induces cortical asymmetry by directly phosphorylating ECT-2 and presented the in vivo phenotypes of the ECT-2 constructs with mutations at the putative phosphorylation sites, which were consistent with their hypothesis.

(Influence of the cortical asymmetry at the mitotic entry)

There is a flaw in their interpretation of the results of various perturbations. Their model in Figure 7B cytokinesis depicts that NMY-2 is absent at the cell cortex during earlier stages of mitosis (pro~meta). This is not precise and misleading. In normal embryos, even after the pseudo-furrowing settles, NMY-2 doesn't disappear from the cell cortex and, importantly, is kept anteriorly enriched though at slightly lower intensity in smaller patches than in the earlier phase (eg. Tse et al. 2012 PMID:22918944, Figure 3). Actin filaments also remain more enriched in the anterior cortex than in the posterior cortex. This is a clear difference from the post-fertilization polarity establishment, in which uniform distribution of the actomyosin network is maintained until the entry of sperm breaks the symmetry.

During the early stages of mitosis, the cortical flow is suspended due to the inhibitory activity of CDK1. The onset of anaphase cancels this inhibition and triggers the contraction of the actomyosin network. If the symmetry of cortical actomyosin is already broken as in the normal embryos, even uniform activation of actomyosin throughout the cell will result in a cortical flow towards the region with a denser actomyosin network (= the anterior cap). If the cortical symmetry is not broken for some reason, it needs to be broken to cause the cortical flow. In theory, the target of symmetry breakage can be any component of the contractile actomyosin network.

The authors observed attenuated ECT-2 asymmetry during cytokinesis in the embryos depleted of SPD-5 (Figure 2C), PAR-3 (Figure 2D), PAR-2 (Figure 2 Supplement 1), NMY-2 (Figure 2 Supplement 1), and AIR-1 (Figure 4A, Figure 4 Supplement 2). In all these cases, the cortical ECT-2 at NEB was found more symmetric (A:P ratio at NEB =1.3, 1.0, 1.2, 1.3, and 1.1, respectively) than the normal embryos (1.4~1.6). In almost all the cases where ECT-2 was asymmetric at NEB (or metaphase), with Galpha(RNAi) as an exception, ECT-2 asymmetry during cytokinesis was normal or enhanced (Tubulin(RNAi)+nocodazole, *nop-1*(it142), zyg-9(b244), dhc-1(RNAi), tpxl-1(RNAi), tpxl-1(RNAi);zyg-9(b244), and saps^-1^(RNAi)). There is a simple and strong correlation between the ECT-2 asymmetry upon mitotic entry and the ECT-2 asymmetry during cytokinesis.

The authors challenge the roles of astral microtubules and dynein in the polar relaxation during cytokinesis based on the observations of the cortical flow in the embryos in which microtubules and spindles are drastically messed up (Tubulin(RNAi)+nocodazole, zyg-9(b244), dhc-1(RNAi)). However, starting with the asymmetric actomyosin network that was successfully established after the fertilization, the anterior-directed cortical flow should occur spontaneously upon reactivation of the contractility following the anaphase onset even without any additional cue. The ECT-2 asymmetry during cytokinesis in these embryos can just be reflecting the fact that these treatments didn't completely disrupt the post-fertilization cortical polarity. Indeed, the ECT-2 asymmetry at the mitotic entry in embryos in Tub(RNAi)+noc and dhc-1(RNAi) embryos was more intense than in the normal embryos (A:P ratio at NEB = 1.9 and 1.8, respectively).

The role of switching of the cortical flow from the P-to-A alone mode to the bidirectional mode in cytokinetic furrow formation has been reported in many papers (PMID: 27719759, 29963981, 32497213, etc.). In this sense, the influence of the anterior centrosome/aster on the anterior cortex is crucial for the spatial control of cytokinesis. This must be a rationale for Mangal 2018 (PMID:29311228) to focus on the role of TPXL-1 in the clearing of anillin from the anterior cortex. Acceleration of furrow formation in correlation with the anterior shift of the anterior centrosome/aster has also been reported (PMID: 32497213). No such test has been performed for ECT-2 localization in this manuscript.

(Phosphorylation of ECT-2 by AIR-1)

Although the authors claim the role of centrosomes based on the spatial correlation, no direct evidence has been provided for the positive role of centrosomes in these embryos. For example, if the ECT-2 asymmetry during anaphase is regulated by centrosomes, disruption of the centrosomes after anaphase onset should disrupt the cortical ECT-2 asymmetry, which is turning over in seconds. In this sense, treatment with MLN8237 (Figure 4-Supplement 1) at metaphase is highly interesting. A caveat here is that MLN8237 is not really specific to Aurora A. It also inhibits Aurora B-INCENP at Ki = 27 nM (just 5~27-fold larger than the Ki for Aurora A) (de Groot et al. 2015). The concentration of MLN8237 used (20 uM) is 700x higher than the Ki for Aurora B-INCENP. The phenotype might be due to the inhibition of Aurora B. Indeed, the ECT-2 signal at the midzone, which is likely to depend on centralspindlin and Aurora B, was lost by the MLN8237 treatment. A mild delay in the removal of ECT-2 from the posterior cortex might have been caused by the inhibition of Aurora B/AIR-2 in addition to or instead of AIR-1.

Point mutations at the phosphorylation sites on ECT-2 are expected to compensate for the above issue of specificity and strengthen the author's theory of direct regulation of ECT-2 by AIR-1. However, as the authors admit in the "Limitations of this study" section, evidence for the in vivo phosphorylation of the putative sites is missing. In addition, it has not been tested whether AIR-1 can phosphorylate these sites. The Glotzer group has revealed key phospho-regulatory mechanisms in cytokinesis (PMID: 15282614, 15854913, 17488623, 19468300). In these works, they showed both the in vivo phosphorylation of the putative phospho-acceptor sites and the in vitro phosphorylation by a protein kinase as well as the in vivo phenotypes of the point mutants of the phosphorylation sites. Currently, what we can conclude from the data in Figures 5 and 6B is that some mutations in a loop in the ECT-2 PH domain mildly affect the cortical association of ECT-2. The gap between this and the phosphorylation of these sites by AIR-1 is huge.

Figure 6 tests AIR-1 depletion in *nop-1*(it142) mutant embryos (A) and the phenotype of the endogenous T634E mutation of ect-2 (B). Is 'Figure 6. AIR-1 is involved in central-spindle-dependent furrowing' (page 42) or 'AIR-1 affects centralspindlin-dependent furrowing' (page 56) an appropriate title for this figure? The dependency on centralspindlin or the central spindle has not been directly tested. Although synthetic defects in *NOP-1* and centralspindlin suppress furrow formation, this doesn't necessarily mean that all the residual cortical activity in *nop-1*(it142) embryos relies on centralspindlin or the central spindle. In *nop-1*(it142) embryos, the NMY-2 cortical accumulation is globally weakened in comparison with the wild-type embryos. Additional depletion of AIR-1 might promote furrowing by facilitating the cortical recruitment of ECT-2 or prevent cytokinesis by suppressing the aster/centrosome-dependent pathway, which may or may not be dependent on centralspindlin. By the way, how does ECT-2 behave in these embryos?

Recommendation for authors:

The feedback loop from myosin to the biochemically upstream regulator RhoGEF is highly intriguing. Focusing on the mechanisms for this phenomenon might be more fruitful than spending time trying to obtain evidence for the phosphorylation on the sites that are not conserved during evolution and only weakly match with the consensus for the Aurora phosphorylation.

(page 17) "As during polarization, basal myosin levels appear to suffice, as ECT-2 asymmetry still increases during anaphase when both *NOP-1* and CYK-4 are inactivated, indicating that bulk, cortical ECT-2 has a low level of RhoGEF activity." Difficult to understand the logical structure due to repeated "as".

Other points

(page 3) "Despite progress in our understanding of cytokinesis, gaps remain in our understanding of the mechanism by which asters spatially regulate RhoA activation." This is correct. However, considering the feedback loop shown by this work (dependence of ECT-2 asymmetry on NMY-2), this might be misleading. The point of regulation of the cortical contraction could be anywhere in the loop (RhoGEF, RhoGAP, Rho, formin, Rho-kinase, myosin phosphatase, myosin-II, actin, actin-bundling proteins, …).

(page 25 method of measuring Boundary Length) "ECT-2:mNG accumulation across all positions along the perimeter of the embryo was fitted to a regression model by GAM (Prediction Accumulation) in R Studio." It is not clear what the 'regression model' was and what the "GAM (Prediction Accumulation) in R Studio". Provide the precise (mathematical) description of the model and the proper reference to the method as well as the R source code.

(page 26 statistical test) It reads that t-test was performed between treatments at every time point. I am not sure whether this is an appropriate approach. Data from different time points from a time series are correlated. It is not clear how this should be reflected in handling the issue of multiple comparisons. Can't we get advice from an expert on statistics?

(Figure Supplement 1 and Figure 5F) Average Accumulation (same as 'Anterior:Posterior Ratio'?) is only shown. Isn't it better to include 'Accumulation on Anterior Cortex' and 'Accumulation on Posterior Cortex' as well for consistency with the other figures?

---

## [Author Response]

1. General Statements

We thank the reviewers for their thoughtful and helpful comments. In general, the reviews were highly positive, although their reviews indicated parts of the manuscript that needed further clarification. We have made extensive changes that improve the clarity and rigor of this submission. We have performed several additional experiments which have extended our analysis in several ways detailed below. None of the conclusions have changed.

The following is a list of eight major changes implemented during the revisions. Point-by-point responses to the reviewers comments follow on subsequent pages.

1. The reviews made clear that we needed to more explicitly discuss the AIR-1 depletion phenotype. This phenotype is complex, it does not result in a complete loss of asymmetry, unlike, for example, depletion of the centrosome component SPD-5. This is because, in AIR-1 depleted embryos, a PAR-2 and cortical flow-dependent pathway induces PAR-2 accumulation at both anterior and posterior poles that induces flows from each pole to the lateral region (Reich 2019, Kapoor 2019, Zhao 2019, Klinkert 2019; PMIDs 31155349, 31636075, 30861375, 30801250). These flows also modulate ECT-2 localization. To clarify this point which came up in multiple reviews, we now include an explanation of the complexity of the AIR-1 phenotype and we present an analysis of ECT-2 localization in embryos depleted of both AIR-1 and PAR-2.

2. In addition to the 95% confidence intervals that were present on our graphs, we now include indications of the results of statistical tests of significance to the results of different treatments.

3. We have revised the analysis ECT-2 accumulation in two ways. First, in the previous draft, we assessed the anterior accumulation over the anterior 40% and the posterior 15% of the embryo. We have revised this analysis comparing the anterior and posterior 20% of the cortex, respectively. This is simpler and more logical in contexts where embryos are symmetric. In addition, we altered the measurements of the length of the posterior boundary. Previously we used a common threshold value, below which we counted pixels to assess boundary length. During the revisions, we noticed that this value was not appropriate for our mutant transgenes which accumulated to higher levels. Therefore, we revised our analysis pipeline such that, for each embryo, we measure the average intensity of the cortex in the anterior 60% of the embryo. We set a threshold of 0.85* this average anterior intensity value. As before, cortical positions below this threshold contribute to the boundary length. This is a more robust and simpler means of evaluating the size of the posterior domain. Neither of these changes affect any of our conclusions, but they are simpler and more rigorous.

4. Most of our figures include quantification of the degree of ECT-2 asymmetry as well as the average anterior and posterior accumulation of ECT-2 as a function of time. While the images show the intensity profiles across the embryo, previously, we did not explicitly show a quantification of the average intensity of ECT-2 as a function of position along the embryo. A new graph, Figure 2Bv, shows this for control embryos and embryos in which tubulin is depleted and depolymerized. This shows that the MT depolymerization results in lower accumulation at the posterior of the embryo and higher accumulation at the anterior.

5. We provide documentary and quantitative evidence that ZYG-9 depletion induces potent cortical flows (Figure 3c and Figure 3, supplement 3), further bolstering the central role of cortical flows in inducing ECT-2 asymmetry.

6. As requested by reviewer 2 (R2b), we have included the analysis of ECT-2 distribution in Gα depleted embryos. As expected due to the lack of spindle elongation, the displacement of ECT-2 from the posterior cortex is greatly attenuated.

7. As requested by reviewer 2 (R2d), we now show that ECT-2C fragments accumulate on the cortex in embryos depleted of ECT-2.

8. One other important point raised by several reviewers concerns the behavior of the ECT-2 T634E allele. This allele, due to the substitution of a phosphomimetic residue, accumulates on the cortex at about 50% the level of the wild-type version. To investigate the possibility that this quantitative difference was the cause of the phenotype, we depleted both the wildtype and mutant ECT-2 constructs by RNAi (these are the sole sources of ECT-2 in the animals). First, we find that wild-type ECT-2 can be depleted to 20% of wild type levels with only a 13% rate of cytokinesis failure (when T634E is depleted to 20%, embryos fail more than 50% of the time). Thus the two-fold reduction in cortical ECT-2 seen in T634E not likely highly significant (ECT-2 is not haploinsufficient). In addition, embryos with ECT-2 T634E initiate ingression in a timely manner, but the furrows ingress more slowly than wildtype. In contrast, depletion of ECT-2 to 20% results in a delay in furrow initiation, but once these furrows form, they ingress at rates similar rates to wild-type. Thus, the T634E variant exhibits a behavior that is quite distinct from that resulting from a (strong) reduction in the levels of wild-type ECT-2.

Reviewer #1 (Evidence, reproducibility and clarity (Required)):SummaryR1a In this study the authors addressed how Ect2 localization is controlled during polarization and cytokinesis in the one-cell *C. elegans* embryo. Ect2 is a central regulator of cortical contractility and its spatial and temporal regulation is of uttermost importance. After fertilization, the centrosome induces removal of Ect2 from the posterior plasma membrane. During cytokinesis Ect2 activity is expected to be high at the cell equator and low at the cell poles. Similarly to polarization, the centrosome provides an inhibitory signal during cytokinesis that clears contractile ring components from the cell poles. Whether and how the centrosomes regulate Ect2 localization is not know and investigated in the study.

This is an accurate summary of the goals of this study.

R1b The authors start by filming endogenously-tagged Ect2 and find that Ect2 localizes asymmetrically, with high anterior and low posterior membrane levels during polarization and cytokinesis. They reveal that the centrosome together with myosin-dependent flows results in asymmetric Ect2 localization. Previous studies had suggested that Air1, clears Ect2 from the posterior during polarization and the authors expand those finding by showing that Air1 function is also required to displace Ect2 from the posterior membrane during cytokinesis.To elucidate if Ect2 displacement is induced by phosphorylation of Ect2 by Air1, the authors investigate the localization of a C-terminal Ect2 fragment containing the membrane binding PH domain. When the predicted Air1 phosphorylation sites are mutated to alanine, the Ect2 fragment still localizes asymmetrically but exhibits increased membrane accumulation.Finally, they investigate the functional role of Air-1 during furrow ingression. They demonstrate that embryos deficient of Air1 and NOP1 have impaired furrow ingression. Lastly, the authors sought to confirm that there is a direct effect of Air1 on Ect2 function by generating a phosphomimetic point mutation of Ect2 using Crispr. They find that the membrane localization of phosphomimetic Ect2 is reduced and consequently furrow ingression is impaired.

This is an accurate summary of our results.

Major commentsR1c It is not convincing that the six putative phosphorylation sites are targeted by the Air1. If Air1 phosphorylation displaces Ect2 from the membrane, a reduction in Ant/Post Ect2 ratio is expected in the phosphodeficient mutants, like after air1 RNAi. However this is not observed for cytokinesis or polarization (Figure 5D(i); E). This suggests that phosphorylation of those sites is not essential for the asymmetric Ect2 localization.

In otherwise wild-type embryos, phosphorylation of these sites is not required for asymmetric ECT-2 localization. Non-phosphorylatable ECT-2 variants exhibit asymmetric localization because these proteins relocalize due to myosin-directed flows. To test the role of phosphorylation, we examine the distribution of ECT-2 and ECT-2C fragments in myosin-depleted embryos in which the flows are blocked, under these conditions, transient local depletion is observed with the phosphorylatable variants, Figure 5E.

While AIR-1 promotes normal polarity establishment, as shown in several recent papers, cortical changes nevertheless occur in the absence of AIR-1. Specifically, a parallel PAR-2 dependent pathway induces weaker flows from both poles toward the equator. To further substantiate the effect of PAR-2 accumulation on ECT-2 accumulation in AIR-1 depleted embryos, we assayed ECT-2 accumulation in air-1(RNAi); par-2(RNAi) embryos (Figure 4, supplement 2). These results show that ECT-2 is nearly symmetric in these double depleted embryos. In addition we have edited the text to describe the unusual bi-polar PAR-2 accumulation that occurs in AIR-1 depleted embryos.

R1d The authors aim to demonstrate that phosphorylation of the identified sites is important for cytokinesis. For this they investigate contractile ring ingression in the phosphomimetic point mutation. Since ring ingression is slower and fails in nop1 mutant they authors conclude that this demonstrates a functional importance of this site. I am not surprised that embryos ingress slower in this mutant since Ect2 localization to the membrane is reduced. This however does not show that this phosphorylation site is the target of the centrosome signal. Importantly, authors would need to demonstrate that Rho signaling and thus Ect2 activity, is increased at the poles, when phosphodeficient Ect2 is the only Ect2 in the embryo.

The fact that a phosphomimetic residue at this site leads to reduced membrane localization is highly relevant, as we suggest that phosphorylation of this site contributes to the mechanism by which AIR-1 generates asymmetric ECT-2. Given the role of AIR-1 in regulating polarity, a version of ECT-2 that can not be phosphorylated would be predicted to be dominant lethal, necessitating a conditional expression strategy which does not currently exist in the early *C. elegans* embryo system (indeed we were unable to recover a T-> A allele at this site, despite extensive efforts). To avoid this issue, we used a viable, fertile, hypomorphic allele that is predicted to be less responsive to AIR-1 activity. The goal of this experiment was to evaluate whether the putative AIR-1 sites affect not only the *NOP-1* pathway for furrow ingression, but also impact furrowing that is centralspindlin-dependent.

To complement this finding have performed experiments in which ECT-2 was partially depleted We used RNAi to partially deplete ECT-2 and ECT-2 T634E and measured the total embryo fluorescence of each ECT-2 variant and the kinetics of furrow ingression. Partial depletion of wt ECT-2, to ~ 20% of control levels leads to delay in furrow formation and all but 2/18 (11%) of embryos complete cell division. In contrast, a similar depletion of ECT-2T634E depletion results in a failure of furrow ingression in ~52 % of embryos. Furthermore, while ECT-2T634E embryos initiate furrowing with normal kinetics, they exhibit a slower rate of furrow ingression, in contrast, partial depletion of WT ECT-2 results in a delay in furrow initiation, but once initiated, the rate of furrow ingression is not significantly affected. These results demonstrate that ECT-2T634E behavior can not simply be explained by a modest reduction in membrane binding.

R1e The authors use the Aurora A inhibitor MLN8237: It was shown prior (De Groot et al., 2015) that this inhibitor is not highly specific for Aurora A, and that it also inhibits Aurora B. Thus experiments need to be repeated with MK5108 or MK8745. They should also be conducted during polarization. Why does Aurora A inhibition not abolish asymmetry? That would be expected?

The role of AIR-1 in symmetry breaking during polarization is previously published, including with chemical inhibitors (Reich 2019, Kapoor 2019, Zhao 2019, Klinkert 2019, PMID 31155349, 31636075, 30861375, 30801250). ECT-2 localization depends on both the spatial regulation of AIR-1 activity and the distribution of cortical factors that contribute to ECT-2 cortical association, as a result of cortical flows. During acute, chemical perturbation of AIR-1 it is likely that these factors, which were polarized prior to drug treatment, remain polarized, allowing the residual cortical ECT-2 to remain asymmetric. The reviewer is correct about the specificity of MLN8237 and we do not rely on it alone to demonstrate the role of AIR-1. Rather this experiment is a complement to our AIR-1 depletion studies, which are sufficient to establish specificity. We present this experiment merely to show that AIR-1 acutely regulates ECT-2 during cytokinesis in embryos that were entirely unperturbed during polarization.

R1f There is no statistical analysis of the results in the entire study. For all claims stating a change in Ant/Post Ect2 ratio or Ect2 membrane localization selected time points should be statistically compared: for example the main point of Figure 1 is that Ect2 becomes more asymmetric during anaphase. Thus a statistical analysis of the Ect2 ratio at anaphase onset (t=0s) and eg. t=90 s after anaphase onset should be performed; or Figure 3A nop-1 mutant Ant/Post Ect2 ratio during polarization: again statistical analysis of control and nop-1 mutant embryos is needed at a particular time point.

All of the graphs were presented with the mean of ~10 embryos per condition and included the 95% confidence intervals. In the revised manuscript, we have included tests of statistical significance, at each time point. While non-overlapping confidence intervals generally suggest statistical significance, we include these analyses on the graphs as it can be difficult to assess statistical significance when the confidence intervals overlap.

R1g The aim of Figure 2B is to demonstrate that Ect2 localization is independent of microtubules, however they still observe some microtubules with the Cherry-tubulin marker and those are even very close to the membrane and therefore could very well influence Ect2 on the membrane. Therefore I am not convinced that this experiment rules out that microtubules have no role in regulating Ect2 localization.

We do not exclude that microtubules play a contributing role in ECT-2 phosphoregulation, but rather we conclude that the primary cue is the centrosome. Indeed, microtubules can play an important role in controlling spindle positioning which affects the proximity of the centrosome to the cortex.

The manuscript states, “Despite significant depletion of tubulin and near complete depolymerization of microtubules (Figure 2B, insets), we observed strong displacement of ECT-2 from a broad region of the posterior cortex during anaphase (Figure 2B).” Thus, despite dramatic reductions in microtubules, not only does ECT-2 become polarized, it becomes hyperpolarized. In contrast, were microtubules directly involved in ECT-2 displacement, one would expect a reduction in polarization as a result microtubule depolymerization. Conversely, though SPD-5 depleted embryos contain far more microtubules than embryos in which microtubule assembly is suppressed, ECT-2 is not polarized in SPD-5 depleted embryos. Thus in the manuscript, we conclude, “Collectively, these studies suggest that ECT-2 asymmetry during anaphase is centrosome-directed.” This conclusion is well supported by the results shown.

R1h Throughout the paper the authors should tone down their statement that Air1 breaks symmetry by phosphorylating Ect2, since phosphorylation of Ect2 by Air2 is not shown.

We agree with this comment and will make the necessary edits to the text. Indeed, this is the reason why we had included the final section in our original draft, “Limitations of this study” which makes this point explicitly.

R1i I understand that the establishment of Ect2 asymmetry is important for polarization. However, how does asymmetric Ect2 localization result in more active Ect2 at the cell equator, which is required for the formation of the active RhoA zone? Would we not expect an accumulation of Ect2 at the cell equator, or if that is not the case more active Ect2 at the equator versus the poles?

The pseudocleavage furrow forms as a result of the anterior enrichment of active RHO-1 and its downstream effectors. There is no evidence for a local accumulation of active RHO-1 specifically at the site of the pseudocleavage furrow. Rather, this furrow forms at the boundary between the portion of the embryo where RHO-1 is active and the posterior of the embryo where RHO-1 is far less active (Figure 1 Supplement 2). We suggest that aster-directed furrowing during cytokinesis likewise results from asymmetric accumulation of the same components, without them necessarily being specifically enriched solely at the furrow.

While cytokinesis generally involves an equatorial contractile ring, furrow formation can be driven by an asymmetric – i.e. non-equatorial – accumulation of actomyosin. This behavior is exemplified during pseudocleavage during which the entire anterior cortex is enriched for actomyosin and the posterior is depleted of myosin (Figure 1 Supplement 2). Several published studies provide evidence that the asymmetric pattern of myosin accumulation contributes to cytokinesis (PMID 22918944, 17669650).

Minor commentsR1j Can the authors explain why the quantification of Ant/Post Ect2 ratio in control embryos differs in different figures? For example: in Figure 1D (i) a slight increase of Ect2 asymmetry ratio is seen at around 80 s after anaphase onset. In comparison, in Figure 2C (i) this increase is not obvious. Are those different genetic backgrounds?

In figure 1 D, time 0 begins at anaphase onset, whereas in 2C, time 0 is specified at the time of nuclear envelope breakdown (NEBD). The duration between NEBD and anaphase onset is ~130 sec and an increase in ECT-2 polarization is observed at 220 s post NEBD, ie 90 sec post anaphase onset comparable to that seen in Figure 1D.

R1k One key point of the paper is that myosin-dependent cortical flows amplify Ect2 asymmetry during polarization and cytokinesis. During polarization the data is convincing, however during cytokinesis Ect2 ratio is only slightly decreased after nmy-2 depletion, again is this decrease even significant?

Figure 3 supplement 1 shows a significant difference in ECT-2 asymmetry between control and myosin-depleted embryos.

R1l In the introduction: "Centralspindlin both induces relief of ECT-2 auto-inhibition and promotes Ect2 recruitment to the plasma membrane" it should be added 'Equatorial' membrane, since Ect2 membrane binding is, to my knowledge, not compromised in centralspindlin mutants or in Ect2 mutants that cannot bind centralspindlin.

Generally speaking, the reviewer is correct that cortical accumulation of ECT-2 globally is centralspindlin independent. However, as seen in e.g. ZYG-9 depleted embryos, ECT-2 is recruited to the posterior cortex in a centralspindlin-dependent manner. Thus centralspindlin can promote ECT-2 accumulation to the cortex and the site of that accumulation will be dictated by the position of the spindle midzone.

R1m Labels in the figures are often very small eg Figure 1 (ii-v) and difficult to read. In addition it is easier for the reader if the proteins shown in the fluorescent images is also labeled in the figure (eg Figure 2B add NG-Ect2).

These useful suggestions have been incorporated.

R1n Material and methods it should be mentioned which IPTG concentration was used.

The IPTG concentration (1 mM) has been added to the revised text.

R1o The authors speculate that the Air1 phosphorylation sites in Ect2 PH domain prevent binding to phospholipid due the negative charge. At the same time, the authors propose that the PH domain binds to a more stable protein on the membrane, which is swept along with the cortical flows and they propose anillin could be that additional binding partner. I might miss something, but do the authors suggest Ect2 has two binding partners: anillin and the phospholipids? It would be necessary to explain this better.The authors should test if anillin represents the suggested myosin II dependent Ect2 anchor. For this they should check if Ect2 localization to the membrane is altered upon on anillin RNAi.

This summary of our model is largely correct, though we do not know the identity of the more stable cortical anchor(s). While we suspect the PH domain binds to a phospholipid, ECT-2 cortical localization also requires ~100 residues C-terminal to the PH domain. It is likely that this domain interacts with a cortical component.

In preliminary experiments, ECT-2 accumulation is not strictly anillin-dependent. However, functional redundancy may obscure a contribution of anillin. Anillin was mentioned simply because of the evidence for a physical interaction between ECT-2 and anillin (Frenete PMID 22514687). In the revised manuscript we also include the possibility that ECT-2 accumulations involves one or more anterior PAR proteins. The identity of the cortical anchor(s) is an interesting question for future studies. We consider this question beyond the scope of the current manuscript.

R1p The title of Figure 3 does not fit the statement the authors want to make, since the key point is how Ect2 polarization is affected and not membrane localization in general.

Thank you for this suggestion. The title has been changed to “Cortical flows contribute to asymmetric cortical accumulation of ECT-2”.

R1q In Figure 4A/C. After air1 depletion the authors observe a reduction in Ect2 asymmetry. Why are the centrosomes not marked in the figures? Because they cannot be detected? The authors would also need to show that the mitotic spindle and centrosomes are no altered by air1 RNAi in the zyg9 mutant. Otherwise the observed effect might be indirect.

Centrosomes are perturbed by depletion of AIR-1 (Hannak, PMID 11748251), but they are still detectable and their positions will be added to figure 4. As has been extensively demonstrated, AIR-1 depletion does lead to attenuated spindles and defects in spindle assembly, some of which are also seen TPXL-1 depleted embryos. These consequences of AIR-1 depletion do complicate the analysis, but this is typical of factors that regulate many processes. This is one of the key reasons why we used ZYG-9 depletion in combination with AIR-1 depletion to overcome these indirect effects.

R1r The authors state that tpxl-1 depletion attenuates Ect2 asymmetry, this is not seen in the quantification (Figure 4Bi). The main phenotype they observe is that Ect2 levels on the membrane increase (Figure 4 ii and iii). They go on testing the function of tpxl1 by depleting tpxl1 in the zyg9 mutant, where the centrosomes are close to the posterior cortex. Here they see no effect on Ect2 asymmetry. Based on that they conclude that tpxl1 has no role in this process. To me this finding is not surprising since the centrosome is close the cortex in zyg9 mutant embryos. Therefore sufficient amounts of active Air1 could reach the membrane and displace Ect2. Thus an amplification of the inhibitory signal by tpxl1 on astral microtubules might not be required. The authors need to mention this possibility and tone down their statment (also in the discussion) that tpxl1 is not required for this process.

In the text, we state, “Cortical ECT-2 accumulation is enhanced by TPXL-1 depletion, though the degree of ECT-2 asymmetry is unaffected (Figure 4B).… we observed robust depletion of ECT-2 at the posterior pole in zyg-9 embryos depleted of TPXL-1, but not AIR-1 (Figure 4C). We conclude that while AIR-1 is a major regulator of the asymmetric accumulation of ECT-2, the TPXL-1/AIR-1 complex does not play a central role in this process.” We consider this to be an accurate description of the results. In sum, we have found no evidence that TPXL-1 contributes to generating ECT-2 asymmetry, beyond its well established role in regulating spindle length and position. The are several other processes that are known to be AIR-1 dependent and TPXL-1 independent; these primarily involve the centrosome (Ozlu, PMID 16054030). Given that TPXL-1 associates with astral microtubules, the fact that microtubule depletion can enhance ECT-2 asymmetry also argues against a requirement for TPXL-1.

R1s It was shown that the C-terminus of Ect2 is sufficient and the PH domain is required for Ect2 membrane localization in *C. elegans* (Chan and Nance, 2013; Gomez-Cavazos et al., 2020). Papers should be cited.

Thank you for this helpful comment. Chan and Nance 2013 indeed shows that the ECT-2 C-term is sufficient to localize to the cell cortex. In contrast, the Gomez-Cavasos paper (PMID 32619481) shows in figure S2 that the PH domain is required for cortical localization of ECT-2; this paper does not focus extensively on cortical accumulation of ECT-2. We have cited Chan and Nance in the revised manuscript.

R1t The authors find that nmy-2 depletion results in loss of asymmetry for the Ect2 C-term and Ect2 3A fragment during polarization. Why is the same experiment not shown for cytokinesis?

Strong depletion of NMY-2 prevents polarity establishment, resulting in symmetric spindles, which in turn results in symmetric ECT-2 accumulation. Thus, the requested experiment would not provide significant additional information.

R1u Air1 is targeted to GFP-C-term Ect2 fragment via GFP-binding to determine the influence on GFP-C-term Ect2 localization (Figure 5F). They state that they see a reduction of Ect2 C-term but not of C-term 3A after targeting. The reader has to compare Figure 5D with F. Since the differences are not big, they need to compare the Ect2 C-term and Ect2 C-term 3A with and without Air1 targeting in the same graph (plus statistics). Otherwise this statement is not convincing.

It is not straightforward to directly compare ECT-2C in the presence and absence of GBP-mCherryAIR-1, because the GBP:AIR-1 fusion protein recruits a large fraction of ECT-2C to the centrosome. For this reason we think it is best to compare the behavior over time of ECT-2C and ECT-2C3A in the presence of GBP-mCherry-AIR-1. At the onset of anaphase, these two fragments localize similarly, but they then diverge over time.

R1v In Figure 6A the authors determine the contribution of air1 to furrowing. For this they deplete air1 in the nop1 mutant. According to previous studies, air1 mutants have a monopolar spindle. How can the authors analyze the function of air1 in cytokinesis when the spindle is monopolar? Did the authors do partial air1 depletion? They authors need to show that there is not major effect on the spindle and centrosome for their conditions. For comparison air1(RNAi) alone has to be included, otherwise the experiment is not conclusive.

AIR-1 depletion does not result in a monopolar spindle in *C. elegans* embryos, though the spindle is attenuated and disorganized (PMID 9778499). TPXL-1 depletion also results in short, well organized spindles (PMID 19889842). The concerns are the reason we performed the ZYG-9 depletion experiments in Figure 4C to ensure the centrosomes are proximal to the cortex.

R1w Upon air1(RNAi) in the nop1 mutant NMY2 intensity seems decreased and not increased. Can the authors comment on that, since that is opposite of what is expected.

This is expected as previous studies have shown that *NOP-1* contributes to RHO-1 activation during polarization and cytokinesis (Tse, PMID 22918944). (NOP stands for No Pseudocleavage).

R1x In Figure 6B they introduce a phosphomimetic point mutation in S634 [sic, T634] in the endogenous Ect2 locus. It not clear why the authors chose this site out of the six putative sites and why they only chose one and not 3 or 6 sites? This needs some explanation.

In our early work with ECT-2 transgenes, we found that a T634E mutation strongly affected cortical ECT-2C, so we decided to assess its affect on the function and localization of endogenous ECT-2. While we were able to recover a T634E variant, we were not able to recover a T634A variant, despite considerable effort. Based on these experiences, we anticipated that we would be unable to recover a mutant version of ECT-2 in which all sites were changed to phosphomimetic.

R1y In the model (Figure 7) no astral microtubules are shown during pronuclear meeting and metaphase. Astral microtubules are present at this stage and should be added to the schematic.

MTs will be added to the figure.

Reviewer #1 (Significance (Required)):R1z. The centrosomes inhibit cortical contractility during polarization and cytokinesis in the one-cell *C. elegans* embryo. Centrosome localized Air1 was proposed to be part of this inhibitory signal, however the phosphorylation target of Air1 is not known. The identification of Ect2 as a phosphorylation target of Air1 would be a great advancement in the field. However, the presented manuscript lacks convincing data that Ect2 is the phosphorylation target of Air1 during polarization and cytokinesis.

We explicitly acknowledge that we have not directly shown that AIR-1 phosphorylates ECT-2. However, we have shown that (i) AIR-1 inhibits cortical ECT-2 localization, (ii) the negative regulator of AIR-1, SAPS^-1^, promotes AIR-1 cortical accumulation, (iii) that the cortical localization domain of ECT-2 has putative AIR-1 sites, which, when mutated to non-phosphorylatable residues leads to increased cortical accumulation of ECT-2 (and (iv) phosphomimetic residues reduce its cortical accumulation), and (v) that these AIR-1 sites are required to render GFP-ECT-2C responsive to GBP-AIR-1. For these reasons we feel that our data makes a strong, albeit indirect, case that AIR-1 regulates ECT-2, even though we clearly acknowledge that we do not directly show that AIR-1 directly phosphorylates ECT-2. Direct proof would require the demonstration that AIR-1 phosphorylates ECT-2 in vivo. This would be difficult to show as ECT-2 phosphorylation is likely transient, it likely affects only a subset of the total ECT-2 pool, and it likely results in loss of membrane association of ECT-2. As it it not possible to synchronize *C. elegans* embryos, biochemical analysis would be very difficult. Even a phosphor-specific antibody for the putative ECT-2 phosphosites might not be particularly informative, as it would be predicted to give a diffuse cytoplasmic signal.

Reviewer #2 (Evidence, reproducibility and clarity (Required)):R2a. In this work, Longhini and Glotzer investigate the localization of an essential regulator of polarity and cytokinesis, RhoGEF ECT-2, in the one-cell *C. elegans* embryo. The authors show that centrosome localized Aurora A kinase (AIR-1 in C. elegans) and myosin-dependent cortical flows are critical in asymmetric ECT-2 accumulation at the membrane. Since membrane interaction of ECT-2 is dependent on the Pleckstrin homology domain present at the Cterminus of ECT-2, they further analyzed the importance of putative AIR-1 consensus sites present in this domain.The authors linked the relevance of these sites in controlling ECT-2 localization and its significance on cytokinesis. The manuscript is well written, the work is interesting, and the data quality is high.

We thank the reviewer for their critique.

Major comments:R2b. In Figure 2, the authors claim that the centrosomes and the position of the mitotic spindle are critical in regulating the asymmetric enrichment of ECT-2 at the membrane. To test the relevance of spindle positioning on ECT-2 localization, the authors depleted PAR-3 and PAR-2. The authors observed that the ECT-2 asymmetry is affected in these settings. However, PAR-3 or PAR-2 depletion impacts polarity, which is critical for many cellular processes, including spindle positioning. Can the authors try to specifically misposition the spindle without affecting polarity? For instance, by depleting Galpha/GPR-1/2 and assessing the impact of such depletion on ECT-2 localization.

Thank reviewer for good suggestion. We have performed the suggested experiment (presented in Figure 2, supplement 2). As one might predict, ECT-2 starts out polarized as Gα is not required for polarity establishment. During anaphase, ECT-2 becomes more symmetric in Gα depleted embryos as compared to wild-type.

R2c. I wonder why the intensity of ECT-2 at the anterior and posterior membrane decreases in air-1(RNAi) post anaphase onset (Figure 4A)? Moreover, I fail to observe a significant asymmetric distribution of ECT-2 in embryos depleted for PERM-1. Therefore it appears that the difference between DMSO and MLN8237-treated embryos is not substantial (at least in the images)?

We do not have a complete or rigorous explanation for all the changes in cortical ECT-2, but they are highly reproducible. We speculate that there are cell cycle regulated changes in ECT-2 accumulation, in addition to its regulation by AIR-1. For example, in figure 1, a strong reduction in both anterior and posterior cortical ECT-2 is evident beginning at approximately -350 sec, which may reflect the initial stages of Cdk1 activation. This may result from cell cycle regulated modulation of ECT-2, as there is evidence that mammalian ECT-2 is subject to a very potent inhibition membrane association by Cdk1 (PMID 22172673). Alternatively, there could be cell cycle modulation of the cortical factor that serves as the “co-anchor” of ECT-2. The ability of GBP-AIR-1 to induce GFP-ECT-2C dissociation also appears cell cycle regulated.

Consistent with a cell cycle regulated component, note that NEBD is delayed in AIR-1 depleted embryos (PMID 17669650, 17419991, 30861375). This delay results in a shorter interval between NEBD and e.g. the peak in Cdk1 activation, explaining the earlier decrease in AIR-1(RNAi) embryos vs. control, relative to NEBD.

Our quantitative analysis indicates a significant increase of cortical ECT-2 upon treatment with MLN8237. In addition, the quantitation in the previous version did show a significant polarization of ECT-2 in PERM-1-depleted embryos prior to treatment. We have revised this figure to simply show an acute increase in cortical ECT-2 upon drug treatment, as the focus of this experiment was solely to show that ECT-2 cortical accumulation is acutely responsive to chemical inhibition during cytokinesis in otherwise normal embryos.

–The data in Figure 5 and 6 are exciting but raise a few concerns:R2d. The authors show that ECT-2C localization mimics the localization of endogenous tagged ECT-2. However, all these analyses with ECT-2C and various mutants are performed in the presence of endogenous ECT-2. Can the author check the localization of these mutant strains in conditions where the endogenous proteins are depleted? I understand that the cortical flow would be perturbed in conditions where endogenous ECT-2 is depleted. However, I suspect that one can analyze the anaphase-specific distribution.

We have examined ECT-2C localization in embryos depleted of ECT-2. Cortical localization of ECT-2C is not dependent upon endogenous ECT-2. This result is now shown in figure 5 supplement 1. However, as the reviewer suggested, embryos depleted of ECT-2 do not show a high degree of ECT-2C asymmetry as ECT-2 is required for the cortical flows that amplify the symmetry breaking during polarization. During cytokinesis, ECT-2C does show a modest change in localization at the poles; the extent of the polar reduction is limited and the changes are symmetric as ECT-2 displacement causes spindles to be symmetrically positioned and limits their elongation during anaphase.

R2e. Can the author comment on why ECT-2C does not accumulate at a similar level as ECT-2C(3A or 6A) at the cell membrane when AIR-1 is depleted (compare Figure 5D with Supplemental Figure 5)?

When ECT-2C(3A or 6A) are expressed in otherwise wild-type embryos, embryo polarization occurs, resulting in anterior-directed flows that concentrate the factor(s) that enables the anterior enrichment of ECT-2 (and ECT-2C 3A/6A). By contrast, when AIR-1 is depleted, most embryos exhibit a “bipolar” phenotype in which PAR-2 is recruited to both anterior and posterior poles, and the actomyosin network becomes somewhat concentrated laterally (PMID 30801250, 30861375, 31636075). The differential positioning of the actomyosin network in AIR-1 depleted embryos is likely responsible for the interesting difference that the reviewer points out. This section of the results states. “Nevertheless, these variants accumulated in an asymmetric manner. ECT-2C asymmetry temporally correlated with anteriorly-directed cortical flows (Figure 5 D,E), raising the possibility that asymmetric accumulation of endogenous ECT-2 drives flows that cause asymmetry of the transgene, irrespective of its phosphorylation status.”

R2f (c). Does the cortical localization of the ECT-2C(6A) mutant become symmetric upon further depletion of AIR-1? Of course, if the asymmetric distribution of ECT-2C(6A) is dependent on the presence of endogenous protein in the cellular milieu, the point raised earlier will help address this concern.

We have not performed this exact experiment with ECT-2C-3A though we have performed it with a longer ECT-2 C-terminal fragment (aa 559-924). As expected, due to the considerations described above, the asymmetry of ECT-2C-3A is reduced when AIR-1 is depleted. Likewise, ECT-2C-6A is becomes symmetric when endogenous ECT-2 is depleted due to the dependence of its asymmetry on cortical flows, as discussed above.

In the revised manuscript, we provide additional explanation of the AIR-1 depletion phenotype which will explain the origin of the asymmetric distribution of ECT-2.

R2g.. The authors predict that the AIR-1 mediated phosphorylation delocalizes ECT-2 from the polar region of the cell cortex. Since the posterior spindle pole is much closer to the posterior cortical region, the delocalization is much more robust at the posterior cell membrane. I wonder why targetting AIR-1 at the membrane (GBP-mCherry-AIR-1) does not entirely abolish GFP-ECT-2C membrane localization? Can the author include the localization of GBPmCherry-AIR-1 in the data? Also, do we know for sure if GBP-mCherry-AIR-1 is kinase active?

The GBP-mCherry-AIR-1 transgene was obtained from the Gönczy lab which demonstrated that it has some activity (PMID 30801250). Given that centrosomal AIR-1 (as compared to astral AIR-1) is the primary pool of AIR-1 responsible for modulating cortical ECT-2 levels, it is a not clear that the GBPfused form of AIR-1 is as active as the centrosomal pool of AIR-1; indeed we suspect it is significantly less active, similar to the manner in which TPXL-1/AIR-1 appears less active towards ECT-2 than centrosomal AIR-1. Indeed as the reviewer suggests, were this pool of AIR-1 highly active, we would expect that its cortical recruitment would preclude embryo polarization, and this transgene would cause lethality when expressed with a GFP-tagged cortical protein. These concerns notwithstanding, we do observe a specific reduction in the anterior accumulation of ECT-2C as compared to ECT-2C3A, suggesting that this form of the kinase has some ability to modulate ECT-2C.

Co-expression of GFP-ECT-2C with GBP-mCherry-AIR-1 induces the centrosomal/astral accumulation of GFP-ECT-2C, which is highly visible in the figure and not seen in the absence of GBP-mCherry-AIR-1. Not surprisingly, the co-expression also induces a cortical pool of GBP-mCherry-AIR-1 that is not seen in the absence of GFP-ECT-2C. These redistributions indicate formation of the complex between GFPECT-2C and GBP-mCherry-AIR-1. The mCherry-AIR-1 images could be added as insets to the figure, but in our opinion, they would not make a substantive contribution, given the dramatic accumulation of centrosomal GFP-ECT-2C.

R2h (e). The authors show that centrosomal enriched AIR-1 [spd-5(RNAi)], but not the astral microtubules localized AIR-1 [tpxl-1(RNAi)], is vital for ECT-2 membrane localization. Interestingly, the authors showed that AIR-1 acts in the centralspindlin-directed furrowing pathway (Figure 6A). I wonder if the authors can combine NOP-1 depletion with TPXL-1 depletion? I guess this will further help to exclude the function of TPXL-1 in the centralspindlin-directed furrowing pathway.

We would like to clarify that our data indicates that AIR-1 acts on both the centralspindlin-independent furrowing (e.g. the anterior furrow in 4C), as well as centralspindlin-dependent furrowing (Figure 6). While the experiment the reviewer proposes appears simple in theory, the interpretation is potentially a bit more complex, due to the role of TPXL-1 in spindle elongation, which can affect centralspindl-indirected furrowing. That said, there are two published experiments and one experiment in the manuscript that indicate that centralspindlin dependent furrowing can occur in TPXL-1 depleted embryos. First, Lewellyn et al. showed that while tpxl-1(RNAi) embryos furrow, tpxl-1(RNAi); zen-4(RNAi) embryos do not, suggesting centralspindlin can function in the absence of TPXL-1. Second, the same paper shows that embryos doubly depleted of TPXL-1 and GPR-1/2 exhibit multiple furrows. Our previous work has shown that furrowing in Galpha-depleted embryos is centralspindlin dependent (Dechant and Glotzer). Furthermore, in the current manuscript we found that embryos depleted of both TPXL-1 and ZYG-9 form posterior furrows (8/8 embryos, 6/8 furrows were strong furrows) although the appearance of these furrows is delayed, presumably due to the reduction in spindle elongation due to TPXL-1-depletion. As described in the manuscript, these posterior furrows have been previously shown to be centralspindlin dependent and *NOP-1* independent.

In accordance with these results, and in direct response to the reviewer’s specific suggestion, we do observe furrowing in *nop-1*(it142); TPXL-1(RNAi) embryos (10/10 embryos furrow, 9/10 complete cytokinesis). Thus, all of the available results indicate that TPXL-1 is largely dispensable for centralspindlin dependent furrowing. However, the role of TPXL-1 in centralspindlin-dependent furrowing is not a focus of the manuscript, thus we do not favor including this result, as it distracts from the primary focus of the study.

R2i (f). Why do NMY-2-GFP cortical levels appear lower in 30% of the embryos that show various degrees of cytokinesis defects (Figure 6A)?

There are a number of possible origins of the variability. As shown in (Reich 2019, Kapoor 2019, Zhao 2019, Klinkert 2019, PMID 31155349, 31636075, 30861375, 30801250), AIR-1 depletion results in variable polarization (unpolarized PAR-2, bipolarized PAR-2, anterior PAR-2, posterior PAR-2). Furthermore, spindles in AIR-1 depleted embryos exhibit somewhat variable positioning. While we were unable to correlate these sources of variability with furrow formation, these results demonstrate that AIR-1 depletion impairs furrowing directed by centralspindlin, which was not entirely expected, given that (i) AIR-1 depletion potently suppresses *NOP-1* dependent flows of cortical myosin, as evidenced by the loss of an anterior furrow in AIR-1(RNAi); *nop-1*(it142) embryos and (ii) centralspindlin directed furrowing can occur in the posterior in ZYG-9 depleted embryos both in the presence or absence of AIR-1 (Figure 4C).

R2j (g). The authors report that phosphomimetic mutation at the phospho-acceptor residue in ECT-2 impacts its cortical accumulation. This strain, together with NOP-1 depletion, affects furrow ingression. One explanation for this phenotype is that phosphomimetic mutant weakly accumulates at the membrane. However, one interesting observation is that ECT-2T634E enriches at the central spindle (Figure 6B, panel 120 sec), which somehow I could not find in the text. Could this additional localization of ECT2 at the central spindle contribute to the cytokinesis defects that the authors have observed? The microscopy images the authors have included show that ECT-2T634E significantly localizes at the equator at the time of furrow initiation. Can the authors add the localization of ECT2 wildtype and ECT-2T634E in NOP-1 depleted conditions where they see an apparent impact on the cytokinesis? Similarly, if the authors include the localization of NMY-2 in these conditions-it will further add more weightage to the data.

We regularly detect trace amounts of ECT-2 on the central spindle and this is slightly enhanced at in the ECT-2T634E mutant. However, given the large cytoplasmic pool of ECT-2, it seems unlikely that the slight enrichment of ECT-2 on the central spindle significantly affects the cortical pool of ECT-2, though the reduction in cortical ECT-2 may facilitate its enrichment on the central spindle.

As shown in figure 3B, depletion of *NOP-1* does not dramatically affect cortical ECT-2 levels in wild-type embryos. Likewise, we did not observe a significant effect of *NOP-1* depletion in ECT-2 T634E, thus we decided not to include this negative result.

As discussed in general point 8, we suggest the modest reduction in the membrane pool of ECT-2 is unlikely to be the primary cause of the T634E, but rather the ability of AIR-1 to modulate induce its relocalization. Consistent with this interpretation, the embryos that failed ingression tended to have more symmetric spindles, which could limit the residual cortical flows that facilitate furrow ingression.

Minor comments:R2k -An explanation of how the timing of NEBD was analyzed in multiple settings would be helpful.

Depending on the experiment, we used either ECT-2:mNG fluorescence (it is excluded from the nucleus until NEBD) and/or the Nomarski images to score NEBD.

R2l -The authors mentioned on p. 6-'Despite significant depletion of tubulin…..during anaphase'. These experiments are performed in the near complete depolymerization of microtubules; thus, regular anaphase will not establish. I understand that the authors are monitoring localization wrt the timing similar to anaphase in the non-perturbed condition, and thus a bit of change in the sentence is required.

Thank you for highlighting this point. We have substituted “following mitotic exit” for “anaphase”. In these images, mitotic exit can be scored by the emergence of contractility.

R2m-After testing the relevance of SPD-5 (that primarily acts on PCM and not on centrioles)-the authors write on p. 6 that 'two classes of explanation…early embryo'. I did not understand the importance of this sentence here.

To clarify, we deleted the words “classes of” from the sentence in question and following that sentence we added the word, “first” indicating that we were explaining the first of the two possible explanations.

R2n-The observed impact of spd-5 (RNAi) on ECT-2 localization could be because of the effects of SPD-5 depletion on centrosomal AIR-1? The authors can link the impact of SPD-5 depletion not only with the centrosome but also with AIR-1 in the discussion.

Indeed, it is well established that SPD-5 is required for centrosomal AIR-1 (Hamill DR, et. Al Dev Cell 2002). The revised discussion now states, “Specifically, during both processes, ECT-2 displacement requires the core centrosomal component SPD-5, which is required to recruit AIR-1 to centrosomes{Hamill et al., 2002, #1201}, but ECT-2 displacement is not inhibited by depolymerization of microtubules and it does not require the AIR-1 activator TPXL-1 (see below).”

R2o-In the various Figure legends, sometimes the authors mention time '0' as anaphase, and other time as anaphase onset.

In all cases, anaphase onset was intended and the legends will be corrected.

Reviewer #2 (Significance (Required)):R2p The manuscript is well written, the work is interesting, and the data quality is of good quality.

We thank the reviewer for their encouragement as well as for their thoughtful critique!

Reviewer #3 (Evidence, reproducibility and clarity (Required)):R3a Symmetry breaking is the process by which uniformity of the system is broken. Many biological systems, such as the body axes establishment and cell divisions in embryos, undergo symmetry breaking to pattern cellular interior design. *C. elegans* zygote has been a classic model system to study the molecular mechanism of symmetry breaking. Previous studies demonstrated critical roles of centrosomes and microtubules in breaking symmetry in the actin cytoskeleton during anterior-posterior polarization and cytokinesis. It, however, remains elusive how centrosomes and/or microtubules regulate the assembly and contractility of the actin cytoskeleton. Recent reports identified Aurora-A AIR-1 as the key centrosomal kinase that suppresses the function of the actin cytoskeleton, but little is known about a substrate of the kinase during symmetry breaking events.Longhini and Glotzer proposed in this manuscript that RhoGEF ECT-2 plays a critical role in symmetry breaking of the actin cytoskeleton under the control of AIR-1 kinase. Kapoor and Kotak (2019) previously proposed the same GEF as a downstream effector of centrosomes, but this work did not provide direct evidence for ECT-2 as the AIR-1 effector. This manuscript identified three putative phospho-acceptor sites in the PH domain of ECT-2 that render ECT-2 responsive to inhibition by AIR-1. Although this manuscript lacks direct in vivo and in vitro evidence for phosphorylation of ECT-2 by AIR-1 kinase, the above findings reasonably support a model where in AIR-1 promotes the local inhibition of ECT-2 on the cortex. Design of the experiments, the quality of images, and data analysis are reasonable, and the main text was written very well. The main conclusion of this work will attract many readers in cell and developmental biology fields. I basically support its publication in the journals supported by Review Commons with minor revisions (see below).

We thank the reviewer for their encouraging remarks and helpful comments.

Minor commentsR3b (1) In Figures 2A and 2B, the authors claimed apparent correlation between spindle rocking and ECT-2 displacement. However, because both MTs and ECT-2 in Fig2AB images are blur, I cannot convince myself whether ECT-2 intensities on the cortex showed negative correlation with the distance between the posterior centrosome and the cortex. The authors may want to provide quantitative data set and use a statistical test to support this conclusion.

Only figure 2A focuses on the rocking. The important structure to assess is the position of the centrosome, as the astral arrays of microtubules are largely radially symmetric (except towards the spindle midzone). As this point in the manuscript were were not discriminating between the astral microtubules and the centrosomes, rather focusing on the overall position of the aster as a whole.

Figures 2B, 2D, Figure 2 Supplements 1 and 2, Figure 3C, and Figure 4B, summarized in figure 7A provide quantitive evidence that the centrosome-cortex distance is an important determinant of ECT-2 cortical accumulation.

R3c (2) Figure 2D would [sic; presumably should] show a ratio between the anterior/posterior pole and the lateral cortex.

The reviewer is presumably noticing that the lateral cortex is brighter than the poles when PAR-3 is depleted. While we agree with this assessment, the point of this experiment was to evaluate whether both centrosomes are equally capable of regulating cortical ECT-2 at the respective poles. It appears to us that comparing the anterior and posterior poles is the appropriate measurement to make to address this point and comparison of the poles to the lateral cortex in par-3(RNAi) vs control would be confusing to readers.

R3d (3) In Figure 3D, the authors need to clarify why they measured ECT-2 dynamics only within the "anterior pole". It would be reasonable to measure ECT-2 dynamics by FRAP and cortical high-speed live imaging on the posterior and the lateral cortex during symmetry breaking.

We measured ECT-2 recovery at a variety of sites with similar recovery kinetics. The comparison of ECT-2 dynamics on anterior and posterior furrows were shown in order to compare ECT-2 dynamics on centralspindlin-dependent and -independent furrows.

We now provide additional supplemental data on ECT-2 dynamics during symmetry breaking. When ECT-2 is polarized, the residual signal is too low to obtain a measure of its recovery.

R3e (4) In Figure 4 supplement, a difference between with or without ML8237 seems marginal. The authors need to show a statistical test to claim "rapid enhancement of cortical ECT-2 after ML8237 treatment".

We will provide a statistical analysis. As the inhibitor affects ECT-2 globally, the anterior/posterior ratio doesn’t change significantly. To avoid confusion, we now present total cortical ECT-2 levels upon anaphase onset in this experiment as this is the most relevant parameter.

R3f (5) I would strongly suggest the authors to clearly state in the first paragraph of discussion that "this working hypothesis is not supported by direct evidence for phosphorylation of ECT-2 by AIR-1 kinase in vitro and in vivo." It should be reasonable to weaken the statement "by Aurora A-dependent phosphorylation of the ECT-2 PH domain" in p13.

We agree with the underlying sentiment (as indicated by the “limitations” section that was present in the original version) and we have revised these sentences accordingly: “Our studies suggest that asymmetric, posteriorly-shifted, spindle triggers an initial focal displacement of ECT-2 from the posterior cortex by Aurora A-dependent phosphorylation of the ECT-2 PH domain, though the evidence for this phosphorylation event is indirect.”

Reviewer #3 (Significance (Required)):See the second paragraph of the Evidence, Reproducibility, and Clarity section.

[Editors' note: further revisions were suggested prior to acceptance, as described below.]

Based on the previous reviews and the revisions, the manuscript has been improved but there are some remaining issues that need to be addressed, as outlined below:The eLife expert raised some points which are transmitted verbatim below. I have read them carefully. I think the vast majority of points can be addressed by rewriting and providing explanations or toning down some of the conclusions. Also, the referee has asked that you provide some more experimental details, statistical methods, and additional citations.Points arising from your response letter and revision:1. In a revised manuscript, I am not convinced with their interpretation of the phenotype of air-1(RNAi);par-2(RNAi) zygotes. Given that either single par-2(RNAi) or single air-1(RNAi) abolished the anterior-enriched distribution of ECT-2::mNG (Figure 4 supplement 2), this data simply indicates the indispensable roles of both PAR-2 and AIR-1 in ECT-2 asymmetry, but they cannot conclude that ECT-2 asymmetry in air-1(RNAi) condition is due to PAR-2-dependent weaker cortical flows. Indeed, the anterior/the posterior ratio of ECT-2::mNG in air-1(RNAi) zygotes shown in Figure 4 supplement 2 is very close to 1.0 throughout mitosis, which is quite different from that in Figure 4A. This discrepancy should be addressed in the final manuscript.

These PAR-2 related experiments were not included in the first version of the manuscript, but they were added in response to the Review Commons referees. As is well documented in several papers (PMIDs 30801250, 30861375, 31636075, 31155349), AIR-1 depletion does not eliminate cortical reorganization, and PAR-2 accumulates, aberrantly, to the cortex in when AIR-1 is depleted. A reproducible accumulation of lateral ECT-2 is observed during mitosis in AIR-1 depleted embryos; which is associated with ectopic furrowing (see point 11, below). While this increase in ECT-2 is visible in the images, it is not surfaced by our standard quantification method which focuses on the anterior and posterior domains. Given this and the established interplay between AIR-1 and PAR-2, we felt it appropriate to include an analysis of ECT-2 localization in embryos deficient in both AIR-1 and PAR-2.

The referee states, “they cannot conclude that ECT-2 asymmetry in air-1(RNAi) condition is due to PAR-2-dependent weaker cortical flows.” We do not discuss PAR-2 in the context of cortical flows in the manuscript. The only sentences in the manuscript that explicitly discuss PAR-2 state ”Likewise, ECT-2 asymmetry during cytokinesis is reduced in embryos depleted of PAR-2.” and “The residual asymmetry in ECT-2 accumulation in AIR-1 depleted embryos is further reduced by co-depletion of PAR-2.”

While there are differences in these datasets, we consider them minor; when the ratios are plotted together only 4/35 timepoints show any degree of statistically significant difference. The pattern of anterior accumulation is virtually identical in the two experiments, but the experiment in the supplemental figure shows a slightly stronger ECT-2 accumulation in the posterior. While the images were processed in the same manner, the two figures were generated with strains with distinct markers: while both figures contain the same ECT-2:mNG allele, in Figure 4A it is paired with with mCh:Tub while Figure 4 supp 2 is paired with NMY-2:mKate, as the two figures were addressing different aspects of the phenotype.

2. The authors' response "While cytokinesis generally involves an equatorial contractile ring, furrow formation can be driven by an asymmetric – i.e. non-equatorial – accumulation of actomyosin. This behavior is exemplified during pseudocleavage during which the entire anterior cortex is enriched for actomyosin and the posterior is depleted of myosin (Figure 1 Supplement 2). Several published studies provide evidence that the asymmetric pattern of myosin accumulation contributes to cytokinesis (PMID 22918944, 17669650)."The role of switching off the cortical flow from the P-to-A alone mode to the bidirectional mode in cytokinetic furrow formation has been reported in many papers (PMID: 27719759, 29963981, 32497213, etc.) as mentioned above. A simple unidirectional asymmetry is not sufficient in discussing the spatial regulation of cytokinesis.

We regret that the context of our answer was apparently not sufficiently clear. The original referee asked “However, how does asymmetric Ect2 localization result in more active Ect2 at the cell equator, which is required for the formation of the active RhoA zone? Would we not expect an accumulation of Ect2 at the cell equator, or if that is not the case more active Ect2 at the equator versus the poles?”

Our initial response was narrowly targeted to this question: “while cytokinesis generally involves an equatorial contractile ring, furrow formation can be driven by an asymmetric – i.e. non-equatorial – accumulation of actomyosin.” Our response implies that while bidirectional mode is the norm, there are cases in which a unilateral mode is sufficient (see bolded words in the response). The pseudocleavage furrow is a prime example of this behavior.

3. The authors' response "However, as seen in e.g. ZYG-9 depleted embryos, ECT-2 is recruited to the posterior cortex in a centralspindlin-dependent manner". I don't understand the logic here. Has this been directly tested by, for example, depletion of ZEN-4 or CYK-4 in zyg-9(b244) mutant embryos? In Figure 3C (zyg-9(b244)), no particular enrichment of ECT-2 was observed at the posterior furrow, which is formed by centralspindlin.

The underlying logic is that the posteriorly positioned, bipolar spindle induces the cortical accumulation of centralspindlin on the adjacent cortex. This pool of centralspindlin apparently recruits ECT-2 from the cytoplasm, despite the strong inhibitory activity of AIR-1 in this region. We infer that this binding may reflect a distinct mode of ECT-2 accumulation, as it appears impervious to AIR-1 activity. Indeed, FRAP experiments indicate this pool of ECT-2 has a larger immobile fraction than the cortical pool of ECT-2 elsewhere in the embryo.

In figure 3D of Werner et al., 2007, we demonstrated that the posterior furrow that forms in embryos with posterior furrows requires centralspindlin. In that particular experiment, MEL-26 depletion was used to induce the spindle to assemble in the posterior. In subsequent unpublished work, we have found that the posterior furrow similarly requires centralspindlin when ZYG-9 depletion is used to reposition the spindle (both MEL-26 and ZYG-9 enhance spindle assembly, albeit via distinct molecular mechanisms).

Additionally, in Tse et al., 2012, we showed that the posterior furrow is largely independent of *NOP-1* and that embryos depleted of both *NOP-1* and CYK-4 fail to furrow altogether.

ECT-2 is readily detected on the posterior furrow once they ingress further than the time point shown in Figure 3C, as shown in Figure 3 Supplement 4.

4. "In contrast, the Gomez-Cavasos paper (PMID 32619481) shows in figure S2 that the PH domain is required for cortical localization of ECT-2; this paper does not focus extensively on the cortical accumulation of ECT-2". I think Gomez-Cavasos should also be cited as these provide complementary information as to the role of the PH domain.

In this case, the focus of this section is which parts of ECT-2 are sufficient for membrane recruitment of ECT-2. That PH domains contribute to membrane accumulation of GEFs has been extensively documented (eg PMID 17007612 from 2006). For example, the PH and the C terminal regions of HsEct2 are required for its membrane accumulation (PMID 27926870). It would not be appropriate to cite Gomez-Cavasos, without citing these papers. The requirement for a PH domain in cortical association does not seem significant enough to warrant a summary of the literature, particularly since membrane binding is the canonical function of PH domains. The somewhat surprising result was that the PH domain was not sufficient, as previously shown by Chan and Nance, which is cited.

5. In addition to the above, please answer either by rewriting or with experiments the points raised by the eLife referee below. As I read it, some experiments on Ect2 localization are required to firmly test your models. All other points raised might involve a balanced discussion of various observations and raising the limitations of the work.eLife referee comments verbatim:Asymmetric actomyosin contractility plays key roles in various cellular activities. In *C. elegans* embryos, both the post-fertilization polarity establishment and cytokinesis depend on this process, which is known to be under the control of Rho GTPase. In this manuscript, the authors studied the molecular mechanism of symmetry breaking, focusing on ECT-2 RhoGEF, the crucial upstream activator of Rho, and its regulation by AIR-1, Aurora A kinase. Although the roles of these molecules both in polarity establishment and cytokinesis have already been reported, it remains unclear whether and how AIR-1 might regulate the activity of ECT-2.By a combination of genetic manipulation and high-quality quantitive microscopy, the authors compared various perturbations and concluded that centrosomes and the cortical flow driven by actomyosin network play roles in the asymmetric cortical localization of ECT-2 while astral microtubules and TPXL-1, a conserved Aurora A regulator that recruits AIR-1 to the astral microtubules, are not essential for the ECT-2 asymmetry during cytokinesis in contrary to the previous reports. Then, the authors tested the hypothesis that AIR-1 induces cortical asymmetry by directly phosphorylating ECT-2 and presented the in vivo phenotypes of the ECT-2 constructs with mutations at the putative phosphorylation sites, which were consistent with their hypothesis.

This referee does not mention that we unambiguously demonstrate that AIR-1 plays an important role in generating cortical asymmetry during cytokinesis. In particular, we show that the ability of centrosomes to inhibit cortical contractility during anaphase depends on AIR-1. Furthermore, we show that AIR-1 is specifically required to induce the displacement of ECT-2 from the cortex during both polarization and cytokinesis. We identify sites on ECT-2 that when mutated to non-phosphorylatable residues increase ECT-2 membrane accumulation one of which, which when mutated to a phosphomimetic residue is sufficient to reduce the cortical accumulation of ECT-2. Further we show that a non-phosphorylatable substitution of these sites impacts the response of ECT-2 C-terminal fragments to an AIR-1:GBP fusion protein. Finally, although AIR-1 clearly regulates the bulk flow of cortical myosin during anaphase, this pool of myosin is largely dependent on a non-essential protein called *NOP-1*. Given the evolutionary novelty of *NOP-1* we further showed that a putative AIR-1 phosphosite also regulates the ability of the conserved cytokinetic regulator centralspindlin to induce cytokinesis.

**Author response image 1. sa2fig1:** Depletion of MRCK-1 does not affect the distribution of cortical ECT-2 during anaphase. MRCK-1 depletion was confirmed by observing the loss of the cortical cap of co-expressed NMY-2:mKate during metaphase.

(Influence of the cortical asymmetry at the mitotic entry)There is a flaw in their interpretation of the results of various perturbations. Their model in Figure 7B cytokinesis depicts that NMY-2 is absent at the cell cortex during earlier stages of mitosis (pro~meta). This is not precise and misleading. In normal embryos, even after the pseudo-furrowing settles, NMY-2 doesn't disappear from the cell cortex and, importantly, is kept anteriorly enriched though at slightly lower intensity in smaller patches than in the earlier phase (eg. Tse et al. 2012 PMID:22918944, Figure 3). Actin filaments also remain more enriched in the anterior cortex than in the posterior cortex. This is a clear difference from the post-fertilization polarity establishment, in which uniform distribution of the actomyosin network is maintained until the entry of sperm breaks the symmetry.During the early stages of mitosis, the cortical flow is suspended due to the inhibitory activity of CDK1. The onset of anaphase cancels this inhibition and triggers the contraction of the actomyosin network. If the symmetry of cortical actomyosin is already broken as in the normal embryos, even uniform activation of actomyosin throughout the cell will result in a cortical flow towards the region with a denser actomyosin network (= the anterior cap). If the cortical symmetry is not broken for some reason, it needs to be broken to cause the cortical flow. In theory, the target of symmetry breakage can be any component of the contractile actomyosin network.

While we dispute that there is a flaw in our interpretations, we do not dispute that NMY-2 does not disappear from the cortex during polarity maintenance, and have revised the model schematic in Figure 7B accordingly. However, our results indicate that this pool of myosin does not play a role in the organization of myosin upon anaphase onset and we have seen no evidence that this anterior cap is sufficient to direct anterior-directed cortical flows. As previously shown (PMID 21737681), polarized myosin clusters assemble normally during anaphase in MRCK-1 depleted embryos. MRCK-1 is known to be required for cortical myosin accumulation during the establishment phase (Figure 6, PMID 19923324).

Notably, the anterior cap of myosin during polarity maintenance (i.e. during mitosis) is qualitatively different from that during interphase. Myosin clusters during maintenance phase are far smaller and less clustered than those present during establishment phase or anaphase. Likewise the organization of actin filaments are quite different during these stages. During maintenance phase Arp-2/3 branched filaments predominate whereas a significant pool of unbranched Formin nucleated filaments assemble during establishment phase and anaphase. These distinct pools of myosin and actin appear to exhibit distinctly different abilities to induce contractions. To substantiate this assertion, we examined the asymmetry of ECT-2 distribution during cytokinesis in control and MRCK-1 depleted embryos. As indicated in the accompanying figure, MRCK-1 depletion has no significant impact on the distribution of ECT-2 during cytokinesis.

This result was predictable. As shown in Figure 2 Supplement 2, although Gα-depleted embryos enter anaphase with asymmetric ECT-2 (and there is no evidence to suggest that Gα-depletion perturbs the myosin cap during mitosis). Despite this preexisting asymmetry, as anaphase initiates, these embryos do not exhibit potent anterior-directed flows that lead to increasingly asymmetric ECT-2, rather ECT-2 becomes progressively more symmetric as anaphase proceeds. The progressive symmetrization of these embryos likely results from the symmetric position of the spindle and its reduced elongation during anaphase. These changes to spindle length and position result in the anterior and posterior centrosomes lying equidistant – and rather distal from the anterior and posterior cortices.

The referees comment “In theory, the target of symmetry breakage can be any component of the contractile actomyosin network” seems to suggest that symmetry breaking is a one time event and that the strength and duration of this symmetry breaking is immaterial. However, as the results of our study shows, dramatically exemplified in Figure 3C and summarized in Figure 7A, the nature of the symmetry breaking events during anaphase have dramatically different consequences on cortical ECT-2 and myosin depending upon the position of centrosomes relative to the cortex. Indeed, figure 2A provides an example of how cortical ECT-2 changes acutely in response to the position of the posterior aster as the spindle rocks. To emphasize this point, we added this sentence to the section of the discussion that focuses on cortical flows, “Further, we speculate that centrosomal AIR-1 not only breaks symmetry, but that this regulation of ECT-2 by centrosomal AIR-1 continues throughout anaphase.”

**Author response image 2. sa2fig2:** Summary figure comparing the average degree of ECT-2 asymmetry at NEBD/Anaphase Onset vs the maximal asymmetry of ECT-2 during anaphase. These values are not correlated.

The authors observed attenuated ECT-2 asymmetry during cytokinesis in the embryos depleted of SPD-5 (Figure 2C), PAR-3 (Figure 2D), PAR-2 (Figure 2 Supplement 1), NMY-2 (Figure 2 Supplement 1), and AIR-1 (Figure 4A, Figure 4 Supplement 2). In all these cases, the cortical ECT-2 at NEB was found more symmetric (A:P ratio at NEB =1.3, 1.0, 1.2, 1.3, and 1.1, respectively) than the normal embryos (1.4~1.6). In almost all the cases where ECT-2 was asymmetric at NEB (or metaphase), with Galpha(RNAi) as an exception, ECT-2 asymmetry during cytokinesis was normal or enhanced (Tubulin(RNAi)+nocodazole, nop-1(it142), zyg-9(b244), dhc-1(RNAi), tpxl-1(RNAi), tpxl-1(RNAi);zyg-9(b244), and saps^-1^(RNAi)). There is a simple and strong correlation between the ECT-2 asymmetry upon mitotic entry and the ECT-2 asymmetry during cytokinesis.

Partial overlap in the requirements for polarization and asymmetric cytokinesis is expected, as these processes rely on a shared machinery. Additionally the degree of asymmetric accumulation of ECT-2 during anaphase depends on the asymmetric positioning of centrosomes, which depend on embryo polarization. In addition, there are exceptions to the correlation the referee cites, and they are highly informative. Gα – depleted embryos polarize normally and enter anaphase with control levels of ECT-2 asymmetry but, upon anaphase onset the degree of the asymmetry of ECT-2 declines. Conversely, the extent of ECT-2 asymmetry prior to anaphase onset in ZYG-9 depleted embryos is similar to wild-type yet during anaphase, ZYG-9-depleted embryos exhibit radically more highly asymmetric ECT-2 during anaphase than control embryos. Contrary to the assertion of the referee, ECT-2 asymmetry during mitosis and its asymmetry during cytokinesis are not well correlated.

The authors challenge the roles of astral microtubules and dynein in the polar relaxation during cytokinesis based on the observations of the cortical flow in the embryos in which microtubules and spindles are drastically messed up (Tubulin(RNAi)+nocodazole, zyg-9(b244), dhc-1(RNAi)). However, starting with the asymmetric actomyosin network that was successfully established after the fertilization, the anterior-directed cortical flow should occur spontaneously upon reactivation of the contractility following the anaphase onset even without any additional cue. The ECT-2 asymmetry during cytokinesis in these embryos can just be reflecting the fact that these treatments didn't completely disrupt the post-fertilization cortical polarity. Indeed, the ECT-2 asymmetry at the mitotic entry in embryos in Tub(RNAi)+noc and dhc-1(RNAi) embryos was more intense than in the normal embryos (A:P ratio at NEB = 1.9 and 1.8, respectively).

See the responses to points 6 and 8 above. Furthermore, “drastically messed up” it is not an accurate description of the spindles in ZYG-9 depleted embryos. These spindles exhibit normal bipolarity and morphology and they segregate chromosomes normally. These spindles are small and, due to a combination of the position of sperm entry and the bias in cortical forces, they assemble close to the posterior cortex.

Were dynein (or microtubules) required for polar relaxation during anaphase, it is reasonable to expect that dynein depletion (or MT disassembly) would result in a reduction in polar relaxation during anaphase, whereas the results show that dynein depletion results in a dramatic increase in polar relaxation.

The role of switching of the cortical flow from the P-to-A alone mode to the bidirectional mode in cytokinetic furrow formation has been reported in many papers (PMID: 27719759, 29963981, 32497213, etc.). In this sense, the influence of the anterior centrosome/aster on the anterior cortex is crucial for the spatial control of cytokinesis. This must be a rationale for Mangal 2018 (PMID:29311228) to focus on the role of TPXL-1 in the clearing of anillin from the anterior cortex. Acceleration of furrow formation in correlation with the anterior shift of the anterior centrosome/aster has also been reported (PMID: 32497213). No such test has been performed for ECT-2 localization in this manuscript.

We concur that the anterior centrosome regulates cortical behavior, including controlling ECT-2 localization. However, because the posterior centrosome is closer to the posterior cortex than their anterior counterparts, the equatorial directed myosin-dependent cortical flows are more pronounced in the posterior domain than in the anterior domain (e.g. Figure 3C). Indeed, the dramatic differences in ECT-2 and NMY-2 localization in ZYG-9 depleted embryos (Figure 3C) is likely to result from the combination of the close proximity of both centrosomes to the posterior cortex and the absence of the anterior centrosome from the anterior domain.

It is important, furthermore, to not only consider the flows of myosin foci that appear, but also the rate at which such foci assemble in the anterior and posterior domains. Two domains can exhibit similar rate of flows, but if the foci are smaller and less numerous in one domain than the other, the net accumulation of myosin in the two domains can differ dramatically.

Importantly, this manuscript focuses on the mechanisms that regulate both the assembly of myosin foci downstream of RHO-1 activation and the subsequent flows (centrosomal AIR-1 inhibiting ECT-2 accumulation), the finding that ECT-2 localization depends upon these flows, and the proposal that asymmetric ECT-2 could function to sustain these flows. These findings extend our current understanding of the mechanism and is consistent with all the data in the literature to our knowledge.

(Phosphorylation of ECT-2 by AIR-1)Although the authors claim the role of centrosomes based on the spatial correlation, no direct evidence has been provided for the positive role of centrosomes in these embryos. For example, if the ECT-2 asymmetry during anaphase is regulated by centrosomes, disruption of the centrosomes after anaphase onset should disrupt the cortical ECT-2 asymmetry, which is turning over in seconds. In this sense, treatment with MLN8237 (Figure 4-Supplement 1) at metaphase is highly interesting. A caveat here is that MLN8237 is not really specific to Aurora A. It also inhibits Aurora B-INCENP at Ki = 27 nM (just 5~27-fold larger than the Ki for Aurora A) (de Groot et al. 2015). The concentration of MLN8237 used (20 uM) is 700x higher than the Ki for Aurora B-INCENP. The phenotype might be due to the inhibition of Aurora B. Indeed, the ECT-2 signal at the midzone, which is likely to depend on centralspindlin and Aurora B, was lost by the MLN8237 treatment. A mild delay in the removal of ECT-2 from the posterior cortex might have been caused by the inhibition of Aurora B/AIR-2 in addition to or instead of AIR-1.

We demonstrate that ECT-2 polarization during cytokinesis depends upon the core centrosomal component SPD-5, though it does not require the astral MTs that emanate from centrosomes.

We have shown the involvement of AIR-1 in ECT-2 regulation by both depletion of AIR-1 and its regulator SAPS^-1^ and a gain of function approach using GBP-AIR-1.

The dramatic difference in ECT-2 localization in embryos deficient in ZYG-9 and AIR-1 as compared to embryos deficient in ZYG-9 and TPXL-1 further supports models in which the centrosomal, I.e. not the astral, pool of AIR-1 is relevant (TPXL-1 is required for AIR-1 to associate with astral microtubules).

The experiments with MLN8237 were merely included to complement the AIR-1 depletion studies and to show that acute inhibition can impact ECT-2 accumulation in otherwise unperturbed embryos. We have no basis to we assert, nor do we assert that these treatments fully inhibit AIR-1 activity. We also do not rule out some affect on AIR-2.

Point mutations at the phosphorylation sites on ECT-2 are expected to compensate for the above issue of specificity and strengthen the author's theory of direct regulation of ECT-2 by AIR-1. However, as the authors admit in the "Limitations of this study" section, evidence for the in vivo phosphorylation of the putative sites is missing. In addition, it has not been tested whether AIR-1 can phosphorylate these sites. The Glotzer group has revealed key phospho-regulatory mechanisms in cytokinesis (PMID: 15282614, 15854913, 17488623, 19468300). In these works, they showed both the in vivo phosphorylation of the putative phospho-acceptor sites and the in vitro phosphorylation by a protein kinase as well as the in vivo phenotypes of the point mutants of the phosphorylation sites. Currently, what we can conclude from the data in Figures 5 and 6B is that some mutations in a loop in the ECT-2 PH domain mildly affect the cortical association of ECT-2. The gap between this and the phosphorylation of these sites by AIR-1 is huge.

The mutational studies on ECT-2 must be considered in context of the loss of function studies of AIR-1 (reduction in AIR-1 activity resulting in increased ECT-2 accumulation; reduction in SAPS-1 activity resulting in increase in AIR-1 activity and an decrease in ECT-2 accumulation. Furthermore, these residues affect the response of an GFP-ECT-2 fusion protein to a GBP-AIR-1 fusion protein). While we would very much like to study the dynamic phosphorylation of ECT-2 in vivo (it is likely highly regulated in space and time) at this moment there are no tools available suitable to do so. The majority of the in vivo phosphorylation studies the referee mentions were performed in bulk lysates from human cells which can be synchronized; this is not possible with *C. elegans* embryos. Further, in the singular case where in vivo phosphorylation was shown in nematode embryos, a phospho-specific antibody was used to label a localized pool of ZEN-4; in this case, ECT-2 phosphorylation by AIR-1 triggers its delocalization. Demonstration of ECT-2 phosphorylation by AIR-1 in vitro would be nice, but neither a positive result nor a negative result would be particularly informative as kinases can be promiscuous in vitro and we can not rule out a requirement for other factors.

Figure 6 tests AIR-1 depletion in nop-1(it142) mutant embryos (A) and the phenotype of the endogenous T634E mutation of ect-2 (B). Is 'Figure 6. AIR-1 is involved in central-spindle-dependent furrowing' (page 42) or 'AIR-1 affects centralspindlin-dependent furrowing' (page 56) an appropriate title for this figure? The dependency on centralspindlin or the central spindle has not been directly tested. Although synthetic defects in NOP-1 and centralspindlin suppress furrow formation, this doesn't necessarily mean that all the residual cortical activity in nop-1(it142) embryos relies on centralspindlin or the central spindle. In nop-1(it142) embryos, the NMY-2 cortical accumulation is globally weakened in comparison with the wild-type embryos. Additional depletion of AIR-1 might promote furrowing by facilitating the cortical recruitment of ECT-2 or prevent cytokinesis by suppressing the aster/centrosome-dependent pathway, which may or may not be dependent on centralspindlin. By the way, how does ECT-2 behave in these embryos?

Furrow formation in wild-type embryos involves both *NOP-1* and centralspindlin. Whereas depletion of *NOP-1* has a dramatic effect on global cortical myosin, it has a limited effect on formation of the contractile ring, which ingresses to completion with near normal kinetics. Conversely, embryos deficient in centralspindlin subunits (CYK-4 or ZEN-4) form slowly ingressing furrows that only partially ingress. Embryos deficient in both *NOP-1* and CYK-4 fail to form furrows. Thus, these two genes function in parallel pathways upstream of ECT-2. These findings are well established in the literature and also shown in figure 3B (PMID 22918944, 26252513, 32619481).

That said, as the referee indicates, depletion of AIR-1 increases cortical ECT-2 and induces a modest degree of hyper-contractility. For example, as shown in Figure 4A, AIR-1 depleted one-cell embryos form multiple furrows during anaphase. Formally, AIR-1 depletion could result in bypass suppression in embryos deficient in either CYK-4 and/or *NOP-1*. However, this hypercontractility depends upon *NOP-1* during pseudocleavage (Figure 5E PMID 31636075) and during cytokinesis as embryos deficient in both AIR-1 and *NOP-1* form a single furrow which is positioned at approximate midplane of the spindle (Figure 6A), suggesting it is centralspindlin directed. Indeed, it was precisely because AIR-1 depletion primarily enhances *NOP-1* dependent contractility, we thought it would be relevant to examine whether AIR-1 has some impact of centralspindlin-directed furrowing. This was the underlying logic for the experiments shown in figure 7.

Nevertheless, to formally rule out the possibility of bypass suppression, we examined furrowing in embryos deficient in AIR-1, *NOP-1*, and CYK-4. Embryos deficient in all three factors (*nop-1*(it142); air-1(RNAi); cyk-4(RNAi)) fail to form furrows during anaphase (and pseudocleavage) (100% of embryos, n=8). The pattern of ECT-2 accumulation…Such embryos exhibit the membrane invaginations characteristic of embryos with weakened cortex (PMID 20808841). There were no significant differences in ECT-2 accumulation in (*nop-1*(it142); air-1(RNAi); cyk-4(RNAi)) embryos as compared to air-1(RNAi) embryos. This sentence was added to the results to reflect these findings, “To test whether AIR-1 depletion does not bypass the requirement for CYK-4 and *NOP-1* in ECT-2 activation, we depleted both AIR-1 and CYK-4 in *nop-1(it142)* embryos. These triply deficient embryos fail to furrow during anaphase (100%, n=8).”

Given the facts above, the finding that AIR-1 depletion and a phosphomimetic substitution of T634 in ECT-2 affects furrowing behavior in *NOP-1* deficient embryos is consistent with AIR-1 involvement in centralspindlin-dependent furrowing and ‘AIR-1 affecting central-spindlindependent furrowing’.

We thank the referee for pointing out the title on page 42, it has been revised to “AIR-1 is involved in centralspindlin-dependent furrowing.”

Regarding the inactivation of *NOP-1* having only a very modest effect on cortical ECT-2 (figure 3B); depletion of AIR-1 results in a significant increase in the accumulation of ECT-2 on the posterior cortex (Figure 4B), and the phosphomimetic substitution of T634 in ECT-2 reduces its cortical accumulation.

Recommendation for authors:The feedback loop from myosin to the biochemically upstream regulator RhoGEF is highly intriguing. Focusing on the mechanisms for this phenomenon might be more fruitful than spending time trying to obtain evidence for the phosphorylation on the sites that are not conserved during evolution and only weakly match with the consensus for the Aurora phosphorylation.

This suggestion is beyond the scope of the current manuscript.

(page 17) "As during polarization, basal myosin levels appear to suffice, as ECT-2 asymmetry still increases during anaphase when both NOP-1 and CYK-4 are inactivated, indicating that bulk, cortical ECT-2 has a low level of RhoGEF activity." Difficult to understand the logical structure due to repeated "as".

Thank you for pointing out this confusing sentence. We have revised it as follows:

“During cytokinesis, basal myosin levels appear to be sufficient to promote asymmetric ECT-2 accumulation, as ECT-2 asymmetry increases during anaphase even when both NOP-1 and CYK-4 are attenuated. We infer that bulk, cortical ECT-2 has a low level of RhoGEF activity."

Other points(page 3) "Despite progress in our understanding of cytokinesis, gaps remain in our understanding of the mechanism by which asters spatially regulate RhoA activation." This is correct. However, considering the feedback loop shown by this work (dependence of ECT-2 asymmetry on NMY-2), this might be misleading. The point of regulation of the cortical contraction could be anywhere in the loop (RhoGEF, RhoGAP, Rho, formin, Rho-kinase, myosin phosphatase, myosin-II, actin, actin-bundling proteins, …).

We have changed the statement to "Despite progress in our understanding of cytokinesis, gaps remain in our understanding of the mechanism by which asters spatially regulate actomyosin contractility.**"**

(page 25 method of measuring Boundary Length) "ECT-2:mNG accumulation across all positions along the perimeter of the embryo was fitted to a regression model by GAM (Prediction Accumulation) in R Studio." It is not clear what the 'regression model' was and what the "GAM (Prediction Accumulation) in R Studio". Provide the precise (mathematical) description of the model and the proper reference to the method as well as the R source code.

We have added the following text to the methods section “ECT-2:mNG accumulation across all positions along the perimeter of the embryo was fitted to a regression model by Generalized Additive Model (GAM) (Prediction Accumulation) in R Studio using the mgcv:gam function.” For additional information the reviewer could refer to https://www.rdocumentation.org/ packages/mgcv/versions/1.8-41/topics/gam

(page 26 statistical test) It reads that t-test was performed between treatments at every time point. I am not sure whether this is an appropriate approach. Data from different time points from a time series are correlated. It is not clear how this should be reflected in handling the issue of multiple comparisons. Can't we get advice from an expert on statistics?

While we are not experts on statistics, we believe that t-test is appropriate. Though data from different time points are correlated, the tests do not involve comparisons of data from neighboring time points. Rather we compare the data at a given time point between conditions. Regarding multiple comparisons, in each case, experimental results are compared to controls, no multiple comparisons are performed (ie we do not compare two different experimental treatments to each other). Finally, in each case, we provide 95% confidence intervals which provide a direct indication of the experimental noise.

(Figure Supplement 1 and Figure 5F) Average Accumulation (same as 'Anterior:Posterior Ratio'?) is only shown. Isn't it better to include 'Accumulation on Anterior Cortex' and 'Accumulation on Posterior Cortex' as well for consistency with the other figures?

Average accumulation is shown in cases where we focus on temporal regulation, as opposed to the spatial regulation. For example, in Figure 1Bi we show average accumulation to demonstrate the overall changes in ECT-2 accumulation during the cell cycle. This is also the case in Figure 4 Supplement 1, where we measure the overall changes in ECT-2:mNG in response to global AIR-1 inhibition with a chemical inhibitor. This is also the case in Figure 5F, where we track the overall accumulation of GFP:ECT-2C and GFP:ECT-2C-3A ,in the presence of GBP:AIR-1, which globally colocalizes with these C-terminal fragments of ECT-2.